# Learning Condensed Graph via Differentiable Atom Mapping for Reaction Yield Prediction

**Ankit Ghosh** [1]   **Gargee Kashyap** [1]   **Sarthak Mittal** [2]   **Nupur Jain** [1]   **Raghavan B Sunoj** [1]   **Abir De** [2]

## Abstract

Yield of chemical reactions generally depends on the activation barrier, i.e., the energy difference between the reactant and the transition state. Computing the transition state from the reactant and product graphs requires prior knowledge of the correct node alignment (i.e., atom mapping), which is not available in yield prediction datasets. In this work, we propose YIELDNET, a neural yield prediction model, which tackles these challenges. Here, we first approximate the atom mapping between the reactants and products using a differentiable node alignment network. We then use this approximate atom mapping to obtain a noisy realization of the condensed graph of reaction (CGR), which is a supergraph encompassing both the reactants and products. This CGR serves as a surrogate for the transition state graph structure. The CGR embeddings of different steps in a multi-step reaction are then passed into a transformer-guided reaction path encoder. Our experiments show that YIELDNET can predict the yield more accurately than the baselines. Furthermore, the model is trained only under the distant supervision of yield values, without requiring fine-grained supervision of atom mapping.

## 1. Introduction

The *yield* of a chemical reaction is expressed as the percentage of conversion of the reactant(s) to product(s). Early prediction of yield before wet-lab validation of reactions can have an immense impact on ML-driven reaction discovery. It allows the identification and removal of low-yielding reactions, thereby helping in the design and optimization of chemical synthesis. Previous works on yield prediction (Ahneman et al., 2018; Nielsen et al., 2018; Zahrt et al., 2019; Sandfort et al., 2020; Schwaller et al., 2020; 2021c; Singh

& Sunoj, 2022; Schleinitz et al., 2022; Lu & Zhang, 2022) have primarily relied on quantum chemically computed molecular descriptors, molecular fingerprints, or SMILES based representation rather than a more holistic representation built on molecular structure itself. In contrast, recent approaches (Saebi et al., 2021; Gong et al., 2021; Kwon et al., 2022; Li et al., 2023) leverage graph neural networks (GNNs) to compute the embedding using molecular graphs.

Yield of a reaction depends strongly on the activation energy, through Arrhenius equation (Arrhenius, 1889), which is associated with the potential energy of a transient chemical entity called the 'transition state'. It is possible to infer the transition state graph from the reactant and product graphs if we know the atom mapping, *i.e.*, the correspondence between the atoms of the reactants and the products (Kim et al., 2024). But most existing yield prediction datasets only include the reactants and products, lacking both transition states and atom mappings. Furthermore, the computation of the true atom mapping from the graph structures of reactants and products is a challenging task. Atom mapping computation involves solving various pairwise graph matching tasks, which are NP hard problems (Astero & Rousu, 2024). In practice, this is usually mitigated by using expert-curated rules. However, such inputs require manual intervention which could, in turn, limit their widespread deployment (Schwaller et al., 2021a). As a result, the existing works on yield prediction do not consider atom mapping or transition states into their model, leaving a significant room for improvement.

### 1.1. Our contributions

We address these challenges by designing a novel yield prediction network (YIELDNET). Specifically, we make the following contributions.

**Differentiable approximation of atom mapping**  In the context of a chemical reaction, atom mapping refers to an alignment map between the reactant and product nodes, which ensures that the atom composition is preserved. Such an alignment can be obtained by solving a graph matching task between the reactants and the products (Körner & Apostolakis, 2008; Astero & Rousu, 2024). However, this necessitates an exploration of a vast permutation space,

---

*Equal contribution  [1]Department of Chemistry, IIT Bombay [2]Department of Computer Science and Engineering, IIT Bombay. Correspondence to: Ankit Ghosh <ankitghosh@iitb.ac.in>.

*Proceedings of the 42nd International Conference on Machine Learning*, Vancouver, Canada. PMLR 267, 2025. Copyright 2025 by the author(s).

which is daunting. To address this challenge, we introduce a fully differentiable node alignment network built on the Gumbel-Sinkhorn iterative procedure. This network takes GNN embeddings of the reactants and products as input and outputs an alignment (doubly stochastic or soft permutation) matrix which approximates the atom mapping.

**Approximate condensed graph of reaction**  We use the atom mapping to compute a condensed graph of reaction (CGR) (Varnek et al., 2005; Nugmanov et al., 2019), as a supergraph containing both reactants and products. Owing to such construction, the CGR serves as a surrogate of the transition state. Subsequently, we feed this CGR into a second GNN to compute its embeddings.

Since the atom mapping is approximated by a neural network, the CGR becomes a trainable weighted graph. Therefore, this second GNN must allow differentiation with respect to the input CGR, which is not supported in standard off-the-shelf GNN models. To address this, we design a GNN that relies exclusively on differentiable operations on the input adjacency matrix, enabling end-to-end training using the CGR.

**Neural reaction encoder**  A reaction may involve multiple sequential steps, often referred to as reaction path. Using the computed representations of the CGR for each step in hand, we compute the representation for the entire reaction path. For each elementary step in a multi-step reaction, we concatenate the embeddings of the CGR with *reaction-step encodings*, akin to positional encodings. Finally, these embeddings are fed into a transformer, and the output from the transformer is then used to compute the reaction yield.

**Learning under distant supervision**  The key goal of YIELDNET is to predict reaction yield. While doing so, it learns an approximate atom mapping using our node alignment network. Notably, YIELDNET is trained solely under the distant supervision of the yield values, without any fine grained supervision of ground truth atom mapping or CGR. Since transition state plays an important role in reaction yield, our approach of using the continuous approximation of the CGR as a surrogate for transition state from the differentiable atom mapping, can enhance the inductive bias of the model.

Our experimental evaluation across multiple datasets show that YIELDNET is able to outperform several baselines by a significant margin. Furthermore, we observe that YIELD-NET can effectively approximate atom mapping under the supervision of only reaction yields.

## 2. Related work

Apart from the aforementioned work on the yield prediction, in recent years, there has been an increasing interest in

designing ML models for atom mapping (Schwaller et al., 2021a; Nugmanov et al., 2022; Astero & Rousu, 2024) and transition state (Pattanaik et al., 2020; Jackson et al., 2021; Makoś et al., 2021; Choi, 2023; Duan et al., 2023; Kim et al., 2024; Duan et al., 2025). However, the problem setup of these works is very different from ours. For instance, the aforementioned works on transition states focus only on predicting the transition states but not yield. To do so, they use the fine-grained supervision of ground truth transition states for training. Similarly, the goal of Nugmanov et al. (2022); Astero & Rousu (2024) is to predict the atom mapping, after training under the supervision of fine-grained ground truth atom mapping. In another work on atom mapping, Schwaller et al. (2021a) employed a large transformer over SMILES representations. In contrast to the above works, our key goal is to predict the yield. Hence, we train our network on datasets containing only yield values, not any ground truth atom mapping or transition states.

Our work is also related to graph neural networks (GNNs), other applications of ML in chemistry, and attention mechanisms in different domains. We briefly review each of them as follows.

**Representation learning for graphs**  Graphs are structured objects that are different from images and texts. They need specialized representation learning methods to compute the graph embeddings. These graph representation learning methods can be broadly divided into two categories. The first set of works consists of transductive models, where the node embeddings are computed independently of each other (Grover & Leskovec, 2016; Perozzi et al., 2014). This requires us to train $O(|V|)$ embeddings separately for each node in a graph. The second set of works consists of inductive models, referred to as graph neural networks or GNNs (Gilmer et al., 2017; Hamilton et al., 2017; Kipf & Welling, 2017; Veličković et al., 2018; Zhang & Chen, 2018), that employ message passing neural networks with shared parameters. At the outset, their goal is to collect information from the neighborhoods at different distances from a node and cast it into a low dimensional representation vector. Such methods are widely used in link prediction (Zhang & Chen, 2018), node classification (Kipf & Welling, 2017), graph matching (Li et al., 2019b; Bai et al., 2019) *etc*.

**ML applications in chemistry**  In recent years, there has been a surge of work that uses machine learning in a wide variety of chemical applications (Hirohara et al., 2018; Bjerrum, 2017; Liu et al., 2018; Huang et al., 2020; Honda et al., 2019; Wang et al., 2019c; Chithrananda et al., 2020). Several works have represented molecules as a customized string called SMILES (Anderson et al., 1987) and then applied CNNs (Hirohara et al., 2018) or sequence encoders (Bjerrum, 2017; Liu et al., 2018; Huang et al., 2020; Honda et al., 2019; Wang et al., 2019c; Chithrananda et al.,

2020). However, the same molecule can result in multiple SMILES strings, which might affect the expressivity of the trained model. Therefore, a wide body of work represents a molecule as a molecular graph and designs graph based machine learning models (Kearnes et al., 2016; Schütt et al., 2018; Qiao et al., 2020; Klicpera et al., 2020b;a; Fuchs et al., 2020; Samanta et al., 2020; Jin et al., 2018; Cao & Kipf, 2018). Such graph based models for molecular graphs can be used for generative modeling (Samanta et al., 2020; Jin et al., 2018; Cao & Kipf, 2018), property predictions (Yang et al., 2019; Axelrod & Gomez-Bombarelli, 2022; Liu et al., 2021; Chithrananda et al., 2020) *etc.*

**Learning alignment in the context of graphs**  Graph matching naturally leads to a quadratic assignment problem (Anstreicher, 2003; Gold & Rangarajan, 1996; Caetano et al., 2009). Fey et al.; Zanfir & Sminchisescu (2018); Wang et al. (2019b) provide a learning model for training graph matching, under distant supervision. In last few years, there are some works on learning node to node alignment for subgraph matching (Roy et al., 2022b; Ramachandran et al., 2024; Raj et al., 2025), maximum common subgraph computation (Kriege et al., 2019; Roy et al., 2022a), graph edit distance (Jain et al., 2024; Li et al., 2019b), graph rerpresentation learning (Roy et al., 2021).

## 3. Preliminaries

**Notations**  We represent a single-step reaction as $A + B \rightarrow Y + Z$, where $R = \{A, B\}$ are the reactants and $I = \{Y, Z\}$ are the products. Here, $I$ typically consists of intermediates in multi-step reactions. Each step involves at most two reactants and two products, consistent with our datasets. For any molecular graph $G_A = (V_A, E_A)$, we use $\boldsymbol{V}_A \in \mathbb{R}^{|V_A| \times d_V}$ and $\boldsymbol{E}_A \in \mathbb{R}^{|E_A| \times d_E}$ to denote the node and edge feature matrices, respectively. In practice, we build the node feature for $u$, i.e., $\boldsymbol{V}_A[u, :]$ using the atomic number, degree of connectivity, hybridization, etc., and the edge feature for $(u, v)$, i.e., $\boldsymbol{E}_A[(u, v), :]$ using the type of bonds respectively. We denote the reactant and product graphs as $G_R = (V_R, E_R)$ and $G_I = (V_I, E_I)$, and their adjacency matrices as $\text{Adj}_R$ and $\text{Adj}_I$, respectively. We denote the set of $N \times N$ permutation matrices as $\mathcal{P}_N$. While $\boldsymbol{P}$ represents both the hard permutation matrix and the doubly stochastic matrix. For a hard permutation matrix $\boldsymbol{P}$, the alignment map $\pi$ satisfies $\boldsymbol{P}[u, u'] = 1 \implies \pi[u] = u'$.

**Reaction and their components**  A reaction path $r$ comprises $n$ elementary steps, with each step represented as $A_i + B_i \rightarrow Y_i + Z_i$, where $Y_i$ and $Z_i$ are intermediates. Inputs $A_i$ and $B_i$ may be initial reactants, catalysts, or intermediates from prior steps ($Y_j$ or $Z_j$, $j < i$). The final product $Y_n$ from the $n$-th step is the reaction's desired output, whose yield is the focus. Thus, $r$ follows a sequential structure:

$$r = \{A_1 + B_1 \rightarrow Y_1 + Z_1,$$
$$\cdots, A_n + B_n \rightarrow Y_n + Z_n\} \qquad (1)$$

At step $i \leq n$, we denote $R_i = \{A_i, B_i\}$ and $I_i = \{Y_i, Z_i\}$. Chemical yield for a reaction can be defined as the observed amount of the desired final product ($Y_n$ in Eq. (1)), which is generally less than the theoretical maximum amount of the product given by stoichiometric calculations.

We do incorporate both catalyst and solvent, when present, into the reactant set $R$. Temperature was excluded as it is invariant in the case of high-throughput experimentation datasets (e.g., DF and NS datasets (Section 5)), which maintain consistent reaction conditions throughout, or varies mildly for others (e.g., SC). Integrating temperature into the node/edge features might enhance CGR or yield prediction quality.

**Activation energy and its relationship with yield**  In an elementary step reaction $R \rightarrow I$, reactants $R$ form a transient and unstable chemical entity (shown as TS in Figure 1), before producing $I$. This chemical entity, called as transition state, has the highest potential energy in the reaction, and the energy required to overcome this barrier is the *activation energy ($\Delta \mathcal{E}^\ddagger$)*. A lower activation energy leads to a faster reaction rate and higher yield (Kozuch & Shaik, 2011).

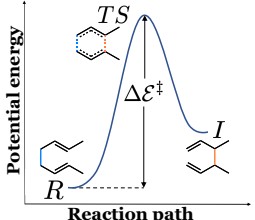

Figure 1: Reaction path vs potential energy

**Atom mapping**  In a chemical reaction, only the bonds in the reactants $R$ are rearranged—broken or newly formed— to produce the products $I$, while the atoms themselves remain conserved. This conservation establishes a correspondence, represented as a permutation $\pi : V_I \rightarrow V_R$, mapping the atoms in the products to those in the reactants. Here, $\pi$ is the atom mapping.

**Our goal**  We are given a set of training dataset $\mathcal{D}$ containing multi-step reactions along with the ground truth yield values, i.e., $\mathcal{D} = \big\{(r_1, \text{yield}(r_1)), ..., (r_{|\mathcal{D}|}, \text{yield}(r_{|\mathcal{D}|}))\big\}$. Any reaction $r$ has $n$ elementary steps as described in Eq. (1), where each elementary step contains a set of two-dimensional representations of the molecular graphs $\{R_i = (A_i, B_i), I_i = (Y_i, Z_i)\}_{i \in [n]}$ during both training and test. Our goal is to develop a yield prediction model that can approximate atom mapping and utilize this mapping to compute a surrogate for the transition state, enabling accurate yield prediction for a new reaction.

## 4. Proposed approach

For brevity of exposition, we first outline a yield prediction model for an ideal setup, where the atom mapping is known.

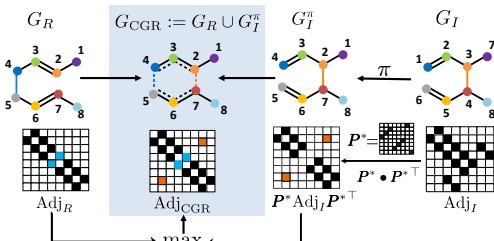

Figure 2: Top row: $G_{\text{CGR}}$ is computed as the supergraph containing $G_R$ and $G_I^\pi \cong G_I$, where $G_I^\pi$ is obtained by applying atom-mapping $\pi$ on $V_I$. Bottom row: Adjacency matrix of CGR is computed as $\max\left(\text{Adj}_R, \boldsymbol{P}^*\text{Adj}_I\boldsymbol{P}^{*\top}\right)$, where $\boldsymbol{P}^*$ is 0/1 permutation matrix corresponding to $\pi$. Given $G_R$ and $G_I$, $\pi$ can be obtained by minimizing the number of edges in the supergraph $G_R \cup G_I^\pi$: this renders in the maximum overlap between $G_R$ and $G_I^\pi$.

Then, we introduce YIELDNET, a neural model that first learns an approximate atom mapping when its ground truth is unavailable, and then computes a condensed graph of reaction (CGR) as a surrogate for the transition state.

### 4.1. Yield prediction when the atom mapping is known

Activation energy directly impacts yield, making it imperative to incorporate transition states in yield prediction models. If the atom mapping $\pi$ is known, we can design a yield prediction model in the following steps, which will enable us to express the yield as a trainable function of the CGR that approximates the corresponding transition state.

**(1)** Given the reactants $R$ and the products $I$ in an elementary step reaction, we permute $V_I$ using the atom mapping $\pi$ to construct $G_I^\pi = (\pi(V_I), E_I(\pi(V_I)))$ which is isomorphic to $G_I$. Then, we compute the graph of transition state as the condensed graph $G_{\text{CGR}} := G_R \cup G_I^\pi$, *i.e.*, a supergraph which subsumes the structures of both $G_R$ and $G_I$ (Figure 2). Suppose $\boldsymbol{P}^* \in \mathcal{P}_N$ is the gold permutation matrix corresponding to $\pi$, where $N = |V_R| = |V_I|$ obtained after padding. This allows us to approximate the adjacency matrix of the CGR using the adjacency matrices $\text{Adj}_R$ and $\text{Adj}_I$ as follows.

$$\text{Adj}_{\text{CGR}} = \max\left(\text{Adj}_R, \boldsymbol{P}^*\text{Adj}_I\boldsymbol{P}^{*\top}\right). \qquad (2)$$

**(2)** We compute the node embeddings of $G_{\text{CGR}_i}$ for each elementary step $i \leq n$ of an $n-$step reaction (1).
**(3)** We feed these node embeddings into a sequence-to-sequence encoder to compute the yield.

### 4.2. YIELDNET model

**Overview**  For most real-life chemical datasets, the atom mapping $\pi$ or the permutation matrix $\boldsymbol{P}^*$ is typically not readily available. Hence, we design neural networks to approximate atom mapping (item (1) above), which leads to an approximate CGR for each reaction step. We achieve this in the following steps, as illustrated in Figure 3.

**(I)** We integrate a graph neural network (GNN) that computes both node embeddings and edge embeddings.
**(II)** We feed these node embeddings into a differentiable node alignment network, Align, to obtain an alignment matrix $\boldsymbol{P} \approx \boldsymbol{P}^*$.
**(III)** To ensure the differentiability of the graph structures of CGR, we use the edge embeddings to obtain a smooth approximation of the adjacency matrix of CGR.
**(IV)** We compute the embeddings of CGR using an input-differentiable GNN.
**(V)** Finally, we combine the embeddings of CGRs of each elementary step using a sequence encoder and obtain the yield.

In the following we describe the above steps in details.

**Embedding computation using GNN**  Given a multistep reaction $r$, each elementary step consists of a reactant graph $G_R$, a product graph $G_I$ and their node and edge features $\boldsymbol{V}_R, \boldsymbol{E}_R$ and $\boldsymbol{V}_I, \boldsymbol{E}_I$, each with $N$ nodes, obtained after padding. We apply a graph neural network $\text{GNN}_\theta$ with parameter $\theta$ on these graphs to compute their node embeddings $\boldsymbol{H}_\bullet$, and edge embeddings $\boldsymbol{M}_\bullet$, as follows:

$$\boldsymbol{H}_R, \boldsymbol{M}_R = \text{GNN}_\theta(G_R, \boldsymbol{V}_R, \boldsymbol{E}_R),$$
$$\boldsymbol{H}_I, \boldsymbol{M}_I = \text{GNN}_\theta(G_I, \boldsymbol{V}_I, \boldsymbol{E}_I). \qquad (3)$$

Here $\boldsymbol{H}_\bullet \in \mathbb{R}^{N \times d}$ and $\boldsymbol{M}_\bullet \in \mathbb{R}^{N \times N \times D}$. For brevity, we present our analysis with $D = 1$ and defer the general discussion using tensor form in Appendix C. To elaborate, for the reactants $R$, $\boldsymbol{H}_R[u,:] = \boldsymbol{h}_R(u) \in \mathbb{R}^d$ is the embedding of node $u$, while $\boldsymbol{M}_R[u,v]$ is the message value $m_R(u,v)$ for an edge $(u,v) \in E_R$ and zero for non-edges $(u,v) \notin E_R$. Similarly, we compute $\boldsymbol{H}_I$ and $\boldsymbol{M}_I$.

**Differentiable approximation of atom mapping**  To address the challenge of learning $\boldsymbol{P}^*$, *i.e.*, the permutation matrix corresponding to the unknown atom mapping, we adopt a data-driven approach. Ideally, $\boldsymbol{P}^*$ should be a binary 0/1 matrix, but such discrete values attenuate gradient signals and prevent backpropagation. To overcome this, we approximate $\boldsymbol{P}^*$ using a node alignment network Align. This network takes the node embeddings of the reactant and product graphs, $\boldsymbol{H}_R$ and $\boldsymbol{H}_I$ as input, and outputs an alignment matrix $\boldsymbol{P}$ which approximates $\boldsymbol{P}^*$. The matrix $\boldsymbol{P}$ is a continuous doubly stochastic matrix, which enables smooth end-to-end optimization.

$$\boldsymbol{P} = \text{Align}(\boldsymbol{H}_R, \boldsymbol{H}_I) \qquad (4)$$
$$= \text{Sinkhorn}\left(-\sum_{\ell=1}^{d}[\max(\boldsymbol{H}_R[u,\ell], \boldsymbol{H}_I[u',\ell])]_{u,u'}\right) \quad (5)$$

Here, $\text{Sinkhorn}(\cdot)$ performs iterative Sinkhorn normalizations on the input matrix (Cuturi, 2013; Mena et al., 2018). Given a temperature $\lambda > 0$, an input matrix $\boldsymbol{C}$ and the number of iterations $T$, $\text{Sinkhorn}(\boldsymbol{C}) = \text{Sinkhorn}^{(T)}(\boldsymbol{C})$

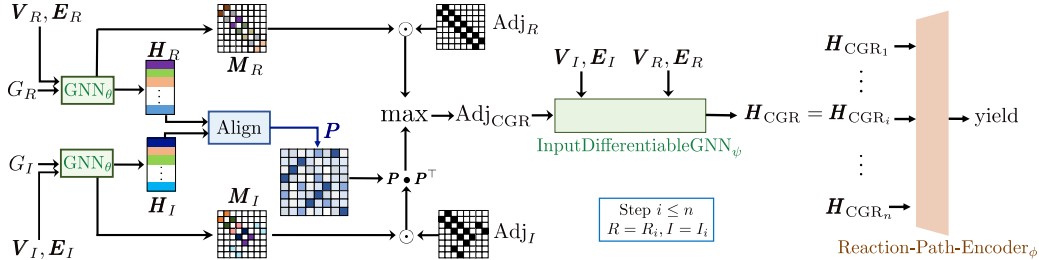

Figure 3: Illustration of YIELDNET. For each reaction step $i$, we process the reactants $R$ and products $I$ using a GNN, $\text{GNN}_\theta$, to obtain node embeddings $\boldsymbol{H}_R, \boldsymbol{H}_I$ and edge embeddings $\boldsymbol{M}_R, \boldsymbol{M}_I$. We feed them into a fully differentiable node alignment network, Align, to obtain an alignment matrix $\boldsymbol{P}$ which approximates the permutation matrix corresponding to the atom mapping. Then, we use $\boldsymbol{P}$ to obtain a continuous approximation of the adjacency matrix of CGR, $\text{Adj}_{\text{CGR}}$. Next, we feed this approximate $\text{Adj}_{\text{CGR}}$ with the node and features into a new GNN ($\text{InputDifferentiableGNN}_\psi$) to obtain $\boldsymbol{H}_{\text{CGR}_i}$, the node embedddings of CGR. Finally, a reaction path encoder use $\{\boldsymbol{H}_{\text{CGR}_i}\}$ to predict the yield.

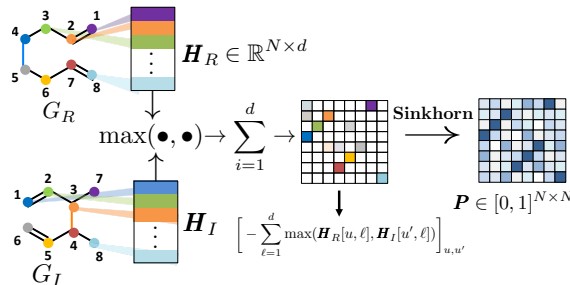

Figure 4: Differentiable approximation of atom mapping using node alignment network Align. It takes $\boldsymbol{H}_R$ and $\boldsymbol{H}_I$, the node embeddings of $G_R$ and $G_I$ as input; and, outputs an alignment matrix $\boldsymbol{P}$, by computing iterative Sinkhorn iterations on the matrix $[-\sum_\ell \max(\boldsymbol{H}_R[u,\ell], \boldsymbol{H}_I[u',\ell])]_{u.u'}$.

is computed as follows:

$$\text{Sinkhorn}^{(0)}(\boldsymbol{C}) = \exp(\boldsymbol{C}/\lambda) \quad (6)$$

$$\text{Sinkhorn}^{(t+1)}(\boldsymbol{C}) = \text{RowDiv}(\text{ColDiv}(\text{Sinkhorn}^{(t)}(\boldsymbol{C}))). \quad (7)$$

Here, RowDiv (ColDiv) indicates division by sum of rows (columns). Note that the limit $\lim_{T\to\infty} \text{Sinkhorn}^{(T)}(\boldsymbol{C})$ converges to a doubly stochastic matrix and the limit $\lim_{\lambda\to 0} \lim_{T\to\infty} \text{Sinkhorn}^{(T)}(\boldsymbol{C})$ converges to a permutation matrix. We run the iterations in Eq. (7) a total of $T = 10$ times.

*Rationale behind our design choice:* Here, we justify the choice of node alignment network in Eq. (5) by connecting this design decision to a combinatorial heuristic for determining atom mapping. From the combinatorial viewpoint, we can compute the permutation matrix $\boldsymbol{P}^*$ corresponding to atom mapping, by minimizing the size of the supergraph containing the structures of $R$ and $I$ (Figure (2)). In terms of the adjacency matrices, this may be written as:

$$\boldsymbol{P}^* = \underset{\boldsymbol{P} \in \mathcal{P}_N}{\arg\min} \sum_{u,v} \max\left(\text{Adj}_R, \boldsymbol{P}\text{Adj}_I\boldsymbol{P}^\top\right)[u,v]. \quad (8)$$

The above optimization problem is a quadratic assignment problem (QAP) due to the quadratic involvement of $\boldsymbol{P}$, and is NP-hard as well. To tackle this challenge, we view a graph

as the "set" of node embeddings and relax the optimization (8) into the problem of minimizing the approximate size of the superset containing $\boldsymbol{H}_R$ and $\boldsymbol{H}_I$. Specifically, we seek to solve: $\min_{\boldsymbol{P} \in \mathcal{P}_N} \sum_{\ell=1}^d \max(\boldsymbol{H}_R, \boldsymbol{P}\boldsymbol{H}_I)[\ell]$. This is same as solving the following optimization task, due to the fact that in each row of $\boldsymbol{P}$, exactly one element equals to one.

$$\min_{\boldsymbol{P} \in \mathcal{P}_N} \sum_{\ell=1}^d \sum_{u,u'} \max\left(\boldsymbol{H}_R[u,\ell], \boldsymbol{H}_I[u',\ell]\right)\boldsymbol{P}[u,u']. \quad (9)$$

The above problem is a linear optimal transport (OT) problem (Kuhn, 1955; Villani, 2008) and is solvable in polynomial time. However, the optimal solution of the above optimization (9) is still a 0/1 hard permutation matrix, which diminishes gradient signals. To address this challenge, we perform a further relaxation of the optimization (9) and solve the following entropy regularized linear OT problem.

$$\min_{\boldsymbol{P}} \sum_{\ell=1}^d \sum_{u,u'} \max\left(\boldsymbol{H}_R[u,\ell], \boldsymbol{H}_I[u',\ell]\right)\boldsymbol{P}[u,u'] - \lambda \cdot \text{Entropy}(\boldsymbol{P}), \quad (10)$$

such that: $\boldsymbol{P} \geq 0, \boldsymbol{P}\mathbf{1} = \boldsymbol{P}^\top\mathbf{1} = 1.$ (11)

Here, $\lambda > 0$ is the regularizer coefficient. As shown by Cuturi (2013); Mena et al. (2018), our alignment matrix $\boldsymbol{P}$ obtained using Eq. (5) is the solution of the above OT problem (10) – (11). Therefore, our node alignment network Align (4) can approximate the true atom mapping, enhancing the overall inductive bias of our model.

**Continuous approximation of CGR** Having computed our alignment matrix $\boldsymbol{P}$ (5), we make a continuous approximation of $\text{Adj}_{\text{CGR}}$ (2), which serves as a surrogate of the transition state of the underlying reaction. This is achieved by performing elementwise multiplication of $\text{Adj}_R$ and $\text{Adj}_I$ with the messages collected from edge embeddings $\boldsymbol{M}_R$ and $\boldsymbol{M}_I$, generated by $\text{GNN}_\theta$ in Eq. (3), *i.e.*,

$$\text{Adj}_{\text{CGR}} \approx \max(\text{Adj}_R \odot \boldsymbol{M}_R, \boldsymbol{P}\text{Adj}_I \odot \boldsymbol{M}_I\boldsymbol{P}^\top) \quad (12)$$

Even without this continuous approximation, the discrete adjacency matrix $\max(\text{Adj}_R, \boldsymbol{P}\text{Adj}_I\boldsymbol{P}^\top)$ is differentiable, since the matrix $\boldsymbol{P}$ is now the output of the node alignment

network. However, there are two key advantages of using the continuous approximation (12).

*(1) $M_\bullet$ can model transient bonds:* CGR represents the transient state, where the edges represent transient fractional bonds. This can be captured better using continuous weights $M$ rather than binary 0/1 values.

*(2) Continuous weights enhance backpropagation:* Binary adjacency matrices suppress gradient signals, limiting model training, whereas the continuous weights help in backpropagation through the GNN. For instance, if instead of $\mathrm{Adj}_I \odot M_I$, we use only the binary adjacency matrix $\mathrm{Adj}_I$, then the gradient term will involve $P\mathrm{Adj}_I \frac{dP^\top}{d\theta} + \frac{dP}{d\theta}\mathrm{Adj}_I P^\top$. Since $P$ involves iterative normalization starting with exponentials, $\frac{dP}{d\theta}$ includes products like $P[u,u']P[v,v']$, due to the fact that $\frac{d}{do_k} e^{o_j}/\sum_i e^{o_i} = -(e^{o_j}/\sum_i e^{o_i})(e^{o_k}/\sum_i e^{o_i})$ when $k \neq j$. Thus, $P\mathrm{Adj}_I\frac{dP^\top}{d\theta}$ contains terms of the form $P[u,u']P[v,v']P[w,w']$, with each $P[.,.] \in [0,1]$, which weakens the gradient signals.

Instead, in our approximation (12), the gradient will involve: $P\mathrm{Adj}_I \odot \frac{dM_I}{d\theta}P^\top + P\mathrm{Adj}_I \odot M_I\frac{dP^\top}{d\theta} + \frac{dP}{d\theta}\mathrm{Adj}_I \odot M_I P^\top$. The first term will include entries like $P[u,u']P[v,v'][\frac{dM_I^\top}{d\theta}][w,w']$. While the last two terms result in the entries like $P[u,u']P[v,v']P[w,w']M[\omega,\omega']$, instead of $P[u,u']P[v,v']P[w,w']$ in binary adjacency matrix representations. These terms with $M$ will enhance the gradient signals.

**Embedding computation for CGR** Next, we compute the node embeddings of the continuous approximation of the CGR, derived in Eq. (12). We first compute the initial node and edge features $V_{\mathrm{CGR}}$ and $E_{\mathrm{CGR}}$, which are required to initialize the GNN embeddings and the message propagation process. These features are composed of two key components. The first component is identical to $V_R$ and $E_R$, *i.e.*, the initial features of the reactants, reflecting the reactants' role in transitioning to the product via the condensed graph. These features are critical as they capture the reactants' structure prior to transformation. The second component is the difference between the features of the reactant set $R$ and the product set $I$, which captures the effect of the reacting atoms and edges. To compute this difference, the nodes and edges of the reactants are aligned with those of the products using the approximate atom mapping $P$ derived from Eq. (5). Therefore, we have:

$$V_{\mathrm{CGR}} = [V_R; V_R - PV_I], \tag{13}$$

$$E_{\mathrm{CGR}} = [E_R; E_R - PE_I P^\top]. \tag{14}$$

Next, we input the features $V_{\mathrm{CGR}}, E_{\mathrm{CGR}}$ and the weighted adjacency matrix $\mathrm{Adj}_{\mathrm{CGR}}$ (12), into a specialized GNN model to compute the node embeddings for the CGR. This process requires the GNN to support backpropagation through both the input adjacency matrix $\mathrm{Adj}_{\mathrm{CGR}}$ and the

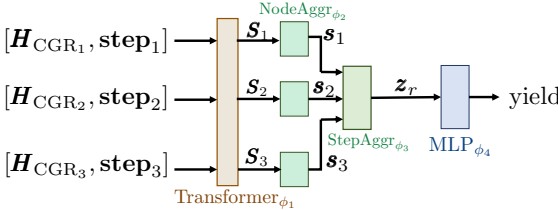

Figure 5: Reaction path encoder. For an $n$-step reaction $r$ with $n = 3$, we encode each step $i \leq n$ by concatenating $H_{\mathrm{CGR}_i}$, the node embeddings of $\mathrm{CGR}_i$, with a step encoding. A transformer then processes this to obtain $S_i$. Next, $S_i$ is aggregated into $s_i$ via $\mathrm{NodeAggr}_{\phi_2}$, and $\{s_i : i \leq n\}$ is passed through $\mathrm{StepAggr}_{\phi_3}$ to compute the reaction embedding $z_r$. We use an MLP on $z_r$ to predict the yield.

features $V_{\mathrm{CGR}}$ and $E_{\mathrm{CGR}}$. However, standard GNN implementations assume fixed, discrete graph structures and do not allow differentiation through the input graph. To overcome this limitation, we design a custom GNN that performs message passing and aggregation using tensorized, differentiable operations on the entries of $\mathrm{Adj}_{\mathrm{CGR}}, V_{\mathrm{CGR}}$, and $E_{\mathrm{CGR}}$. We denote this GNN as $\mathrm{InputDifferentiableGNN}_\psi$, parameterized by $\psi$, which computes the embeddings $H_{\mathrm{CGR}} \in \mathbb{R}^{N \times d_H}$ as shown below. Appendix C contains more details.

$$H_{\mathrm{CGR}} = \mathrm{InputDifferentiableGNN}_\psi(\mathrm{Adj}_{\mathrm{CGR}}, V_{\mathrm{CGR}}, E_{\mathrm{CGR}}) \tag{15}$$

**Reaction path encoder** For a multistep reaction $r$, the elementary steps occur in a unique sequence. Therefore, encoding the reaction path requires that CGR of each step should be associated with a signal that captures its position in the reaction path. However, the node embeddings $H_{\mathrm{CGR}}$ for the different steps are independent of the ordering of the steps. Therefore, to embed condensed graphs in a sequence-aware manner, we concatenate $H_{\mathrm{CGR}}$ with *step-encodings* similar to positional embeddings and pass them through a layer of transformer. Given a $n$-step reaction $r$, we first compute the sequence-aware node embeddings $S_i$:

$$\overline{H}_i = [H_{\mathrm{CGR}_i}, \mathbf{step}_i], \quad i \in \{1, .., n\} \tag{16}$$

$$S_1, ..., S_n = \mathrm{Transformer}_{\phi_1}(\overline{H}_1, ..., \overline{H}_n.) \tag{17}$$

Here, $\mathrm{CGR}_i$ is the CGR for the $i$-th elementary step. Next, we aggregate $S_i$ to their graph level embedding $s_i$ using a $\mathrm{NodeAggr}$, which is a permutation invariant set encoder (Vinyals et al., 2016). We compute $s_i$ as:

$$s_i = \mathrm{NodeAggr}_{\phi_2}(S_i[1,:], ..., S_i[N,:]), \quad \text{for } i \leq n. \tag{18}$$

We could alternatively apply $\mathrm{NodeAggr}$ to $H_{\mathrm{CGR}_i}$ first, followed by the $\mathrm{Transformer}$. However, this approach would aggregate the node embeddings of a CGR into its graph embedding before introducing interactions. Hence, this would only allow interactions between CGRs at a more coarse level. In contrast, applying the $\mathrm{Transformer}$ before $\mathrm{NodeAggr}$ allows the model to capture finer interactions between atoms within the CGRs across different steps.

Next, we aggregate all the graph embeddings $\{s_i : i \leq n\}$

to obtain the reaction embedding $z_r$ using $\text{StepAggr}_{\phi_3}$, which is another set encoder of the same architecture as proposed by (Vinyals et al., 2016). Finally, we feed the $z_r$ into a multilayer perceptron (MLP) to predict reaction yield value for the underlying multistep reaction $r$.

$$z_r = \text{StepAggr}_{\phi_3}(s_1, ..., s_n) \qquad (19)$$

$$\text{yield} = \text{MLP}_{\phi_4}(z_r). \qquad (20)$$

Here, the set $\phi = \{\phi_1, \phi_2, \phi_3, \phi_4\}$ is the collection of all trainable parameters.

### 4.3. Training

We conduct end-to-end training of the entire network to minimize the mean squared error (MSE) between the predicted and gold yield values, which serves as the loss function. Note that our node alignment network does not explicitly ensure that atoms of the reactants only match with the same atoms in the products, e.g., it does not ensure carbon matches with carbon and not oxygen. We ensure this goal by explicit regularization described as follows.

**Regularizer to constrain atom mapping** We introduce a regularizer that penalizes the $P[u, v]$ values for those $(u, v)$ pairs where $u$ and $v$ have different atomic identities (or, atomic numbers) since an atom doesn't convert to another atom throughout the reaction. Specifically, we introduce a matrix $\Lambda \in \mathbb{R}^{N \times N}$ where $\Lambda[u, v] = \mathbb{I}(\text{atom}_R(u) \neq \text{atom}_I(v))$ and $\mathbb{I}(\cdot)$ is the indicator function. Then, we compute the regularizer $\text{Reg}(R, I) = ||P \odot \Lambda||_F^2$ for each CGR to limit the exploration space of $P$, where $P$ is computed in Eq. (5).

**Loss** Given a set of reactions $\mathcal{D} = \left\{ (r_1, \text{yield}(r_1)), ..., (r_{|\mathcal{D}|}, \text{yield}(r_{|\mathcal{D}|})) \right\}$, we minimize the sum of the squared error between predicted yield, $\text{yield}_{\theta, \psi, \phi}(r)$ computed using Eq. (20) and the ground truth yield $\text{yield}(r)$, in the presence of the atom mapping regularizer as described above. Given a regularization parameter $\rho$, the training optimization becomes:

$$\min_{\theta, \psi, \phi} \sum_{r \in \mathcal{D}} \left[ (\text{yield}_{\theta, \psi, \phi}(r) - \text{yield}(r))^2 + \rho \sum_{(R, I) \in r} \text{Reg}(R, I) \right] \qquad (21)$$

## 5. Experiments

In this section, we provide a comprehensive evaluation of our method across eight datasets. Appendix E contains additional experiments and results on more datasets including USPTO dataset (Lowe, 2017). Our code is available in https://github.com/ankitthreo/YieldNet.git.

### 5.1. Experimental setup

**Datasets** We carry out our experiments using eight datasets. They include – (1) GP dataset which is derived from Gasphase Isomerization reactions (Grambow et al., 2020b); five datasets derived from catalytic asymmetric N, S-acetal

formation reaction (Zahrt et al., 2019), *viz.*, (2) NS1, (3) NS2, (4) NS3, (5) NS4, (6) NS5; one dataset on (7) Suzuki coupling reaction (SC); and, another dataset on (8) Deoxyflurorination (Nielsen et al., 2018) (DF). GP is the simplest dataset, where each reaction is single-step reaction and the ground truth atom mapping is available. The remaining seven datasets involve multi-step reactions and *do not* contain any ground truth atom mapping. Availability of atom mapping in GP helps us to better evaluate our atom mapping approximator and understand the effect of CGR on yield prediction, which is not possible for the other more commonplace datasets. Appendix D contains more details about the datasets.

**Baselines** We evaluate our model against several baselines including (1) YieldBERT (Schwaller et al., 2021c), (2) DeepReac+ (Gong et al., 2021) and several variants of graph neural networks: (3) graph convolutional network (GCN) (Kipf & Welling, 2017), (4) heterogenous graph transformer (HGT) (Hu et al., 2020), (5) topology adaptive graph convolutional networks (TAG) (Du et al., 2017) and (6) graph isomorphism network (GIN) (Xu et al.).

**Evaluation metrics** We partitioned the datasets into 70% training, 10% validation, and 20% test folds. We generated ten random splits using different random seeds. For each split, we measure the performance in terms of mean absolute error (MAE) between the predicted and ground truth yield values on the test set, and then average it across all ten splits to report the overall performance.

### 5.2. Results

**Comparison with yield predictor baselines** First, we compare YIELDNET against the baselines across eight datasets. We also report on a skyline variant (YIELDNET (sky)), which predicts yield using the true CGR, instead of an approximation. Since true CGR is only available in GP, skyline performance is limited to only GP dataset. Table 1 shows the results in terms of MAE. We make the following observations. (1) YIELDNET (sky) outperforms YIELD-NET, highlighting CGR's importance in yield prediction. (2) YIELDNET outperforms all baselines in seven out of eight datasets; and, in five datasets, the performance gains are statistically significant. The baselines do not explicitly approximate CGR and instead use the embeddings of the reactants and products as-they-are, leading to weaker performance. (3) There is no consistent second-best model; GCN, TAG, DeepReac+, and YieldBERT share the second-best across datasets.

**Evaluation of our atom mapping approximation** Next, we compare our approximate atom mapping computed using Eq. (5), with four existing atom mapping methods. They include (1) RDKit (Landrum, 2013): the alignment strategy built into the RDKit library, (2) RXNMapper: a pre-trained

| Model | GP | NS1 | NS2 | NS3 | NS4 | NS5 | SC | DF |
|-------|-----|-----|-----|-----|-----|-----|-----|-----|
| GCN | $38.057 \pm 0.337$ | $11.162 \pm 0.735$ | $9.416 \pm 1.045$ | $8.813 \pm 0.857$ | $8.777 \pm 0.221$ | $4.453 \pm 0.293^*$ | $10.682 \pm 0.396$ | $12.516 \pm 0.329$ |
| HGT | $39.405 \pm 0.231$ | $13.385 \pm 0.844$ | $9.573 \pm 1.016$ | $9.421 \pm 0.835$ | $8.629 \pm 0.249$ | $4.561 \pm 0.277$ | $13.665 \pm 0.334$ | $19.500 \pm 0.257$ |
| TAG | $36.887 \pm 0.412$ | $13.246 \pm 0.760$ | $9.560 \pm 0.994$ | $9.232 \pm 0.860$ | $8.603 \pm 0.230$ | $4.547 \pm 0.286$ | $12.603 \pm 0.450$ | $18.150 \pm 0.447$ |
| GIN | $38.307 \pm 0.450$ | $13.318 \pm 0.825$ | $9.588 \pm 1.017$ | $9.152 \pm 0.860$ | $8.617 \pm 0.217$ | $4.568 \pm 0.285$ | $12.899 \pm 0.389$ | $17.126 \pm 0.479$ |
| DeepReac+ | $27.837 \pm 0.346$ | $12.985 \pm 1.452$ | $10.729 \pm 0.690$ | $9.968 \pm 0.981^*$ | $9.439 \pm 0.775^*$ | $5.125 \pm 0.301^*$ | $16.953 \pm 1.800$ | $14.022 \pm 1.694$ |
| YieldBERT | $40.910 \pm 0.300$ | $12.446 \pm 0.820$ | $9.593 \pm 1.002$ | $8.780 \pm 0.918^*$ | $9.236 \pm 0.290$ | $4.532 \pm 0.255$ | $10.437 \pm 0.328$ | $12.470 \pm 0.412$ |
| YIELDNET | $23.152 \pm 0.393$ | $9.245 \pm 0.518$ | $8.387 \pm 0.907$ | $7.914 \pm 0.931$ | $7.015 \pm 0.495$ | $4.382 \pm 0.249$ | $8.751 \pm 0.438$ | $6.941 \pm 0.192$ |
| YIELDNET (sky) | $17.396 \pm 0.357$ | NA | NA | NA | NA | NA | NA | NA |

Table 1: Comparison of yield prediction performance for YIELDNET against all the competitive baselines, *viz.*, GCN (Kipf & Welling, 2017), HGT (Hu et al., 2020), TAG (Du et al., 2017), GIN (Xu et al.), DeepReac+ (Gong et al., 2021), YieldBERT (Schwaller et al., 2021c), on the 20% test examples, across all datasets. Performance is measured in terms of Mean Absolute Error (MAE). YIELDNET(sky) represents a skyline of our model, where we use the true CGR. Only GP dataset contains the true CGR and therefore, such skyline performance is not available for other datasets. Numbers in green (yellow) indicate the best (second best) performer. Our improvement in performance over the next best baseline, where YIELDNET is the best performer, is statistically significant with p-value $< 0.05$, except in the cases marked with $^*$.

| Model | NS1 | NS2 | NS3 | SC |
|-------|-----|-----|-----|-----|
| RDKit | 9.910 | 8.845 | 8.567 | – |
| RXNMapper | 9.610 | 8.871 | 8.470 | 10.046 |
| Random | 11.195 | 8.984 | 8.695 | 10.223 |
| Attention | 9.548 | 9.024 | 8.432 | 8.826 |
| YIELDNET | 9.245 | 8.387 | 7.914 | 8.751 |

Table 2: Comparison between different atom mapping strategies in terms of MAE of yield prediction

node alignment model (Schwaller et al., 2021a) on a large set of reactions, (3) Random: Uniformly sampled alignment, and (4) Attention: an attention network between reactants and products based on Astero & Rousu (2024), that induces a non-injective alignment, unlike our proposed alignment network which outputs an approximate injective alignment. Here, RDKit requires a well-defined reaction template obtained through expert-curated rules and subsequent manual inspection of atom mapping for each reaction in the dataset. Consequently, we cannot generate such templates for SC datasets due to the high diversity of reactions. In Table 2, we report the results in terms of the MAE of the predicted yield. We make the following observations: (1) Our method outperforms the alternatives. (2) The attention network outperforms RXNMapper, RDKit and Random in most cases. However, since it induces a non-injective mapping, it assigns a product atom to multiple reactant atoms with high probability. As a result, our method outperforms it, since our alignment network provides the atom mapping through a doubly stochastic matrix, which in turn induces an injective mapping (one atom in $R$ is mapped to exactly one atom $I$ and vice-versa). (3) The random permutation generation strategy performs the worst. This underscores the necessity of good approximation of atom mapping.

Next, we evaluate the accuracy of our learned atom mapping by measuring its proximity to the ground-truth atom mapping available in the GP dataset— the only dataset that provides such ground-truth mappings. Specifically, we compute the average Frobenius norm of $||\boldsymbol{P} - \boldsymbol{P}^*||_F$ across all reactions, where $\boldsymbol{P}^*$ is the node permutation matrix representing the ground-truth atom mapping and $\boldsymbol{P}$ is the learned node

alignment matrix. We compare our method with attention-based method (Astero & Rousu, 2024), which was the best alternative to our model, as shown in Table 2. Our method gives the average error using $||\boldsymbol{P} - \boldsymbol{P}^*||_F = \mathbf{3.708}$, whereas attenion based method gives $||\boldsymbol{P} - \boldsymbol{P}^*||_F = 5.096$. Notably, our method achieves such high accuracy in atom mapping prediction solely through end-to-end learning from yield data, without any access to the ground-truth atom mapping.

**Interplay of atom mapping with predicted yield** Here, we examine the extent to which the performance of YIELDNET can be attributed to its ability to learn the atom mapping. We investigate the relationship between the mismatch in prediction error $\Delta \text{AE}_r$, and the difference between the true and learned atom mapping: $\Delta \boldsymbol{P}_r$ for each test reaction $r$ as follows. We consider GP datasets since it is the only dataset containing atom mapping. Our approach begins by training a

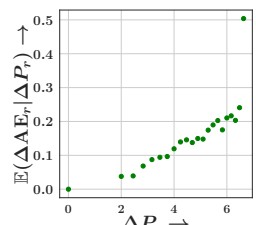

Figure 6: $\Delta \text{AE}_r$ vs $\Delta \boldsymbol{P}_r$

different variant of our model where we replace $\boldsymbol{P}_r$ for each reaction $r$ with $\boldsymbol{P}_r^*$, *i.e.*, the permutation matrix corresponding to the ground truth atom mapping. We call this model as *reference model*. Then, for the test dataset, we gather all the learned alignments $\boldsymbol{P}_r$ from our trained model $\text{yield}_{\theta,\psi,\phi}$ and transform them into hard permutation matrices $\boldsymbol{P}_{\text{hard},r}$ via the Hungarian algorithm. Next, we apply both $\boldsymbol{P}_r^*$ and $\boldsymbol{P}_{\text{hard},r}$ to the reference model to obtain the predictions $\text{yield}_{\boldsymbol{P}_r^*}(r)$ and $\text{yield}_{\boldsymbol{P}_{\text{hard},r}}(r)$ for each reaction $r$ in the test set. Note that $\text{yield}_{\boldsymbol{P}_r^*}$ gives high accuracy as we leak the true atom mapping into the model. Next, we calculate the difference in the absolute errors (AE) for each test reaction as $\Delta \text{AE}_r = |\text{AE}_r^* - \text{AE}_{\text{hard},r}|$ where $\text{AE}_r^* = |\text{yield}_{\boldsymbol{P}_r^*}(r) - \text{yield}(r)|$ and $\text{AE}_{\text{hard},r} = |\text{yield}_{\boldsymbol{P}=\boldsymbol{P}_{\text{hard},r}}(r) - \text{yield}(r)|$ for the gold yield values $\text{yield}(r)$. Finally, we quantify the dissimilarity between two permutation matrices using $\Delta \boldsymbol{P}_r = ||\boldsymbol{P}_r^* - \boldsymbol{P}_{\text{hard},r}||_F$

and plot the correlation between $\mathbb{E}(\Delta\mathrm{AE}_r|\Delta\boldsymbol{P}_r)$ and $\Delta\boldsymbol{P}_r$, by averaging $\Delta\mathrm{AE}_r$ at particular $\Delta\boldsymbol{P}_r$. Figure 6 shows a strong correlation between $\Delta\boldsymbol{P}_r$ and $\Delta\mathrm{AE}_r$, indicating a strong association between the learned alignments' quality and the model's predictive performance.

**Ablation study on CGR representations**  Here, instead of incorporating the embeddings of the CGR into the reaction path encoder, we directly feed the embeddings of the reactants $R_i$ and the products $I_i$ into the encoder. Specifically, we omit both our node alignment network Align and CGR representation network InputDifferentiableGNN, and instead input the embeddings from the GNN into the transformer, alongside the step encoder. Table 3 summarizes the results. We observe that incorporating CGR embeddings into the reaction path encoder is more effective than directly utilizing the embeddings of the reactants and products.

| Method | NS1 | NS2 | NS3 | SC |
|---|---|---|---|---|
| w/o CGR | 11.356 | 9.459 | 9.342 | 10.686 |
| YIELDNET | 9.245 | 8.387 | 7.914 | 8.751 |

Table 3: Effect of ablation of CGR (MAE).

**Ablation study on reaction encoder components**  Here, we investigate the benefits offered by different components of our reaction path encoder (Eq. (16)—Eq. (20)). Specifically, we focus on ablations of its two key components, *viz.*, the transformer (17) and StepAggr (19). Table 4 summarizes the results for four multi-step reaction datasets measured in terms of MAE of the predicted yield. We make the following observations: (1) the transformer encoder contributes to the improved accuracy across most cases; (2) utilizing Seq2Seq for sets (Vinyals et al., 2016) yields better performance than DeepSet (Zaheer et al., 2017) and simple sum aggregation methods; (3) ablation on the set encoder has a stronger impact on MAE than the transformer because the set encoder is applied in the final layer.

| Method | NS1 | NS2 | NS3 | SC |
|---|---|---|---|---|
| StepAggr = DeepSet | 9.643 | 9.008 | 8.283 | 8.551 |
| StepAggr = SumAggr | 10.192 | 8.858 | 8.349 | 8.853 |
| Without transformer | 9.467 | 8.680 | 8.164 | 8.628 |
| YIELDNET | 9.245 | 8.387 | 7.914 | 8.751 |

Table 4: Effect of Ablation on reaction path encoder components (MAE).

**Effect of ablation on the regularizer Reg$(R, I)$**  Here, we evaluate the effect of the regularizer Reg$(R, I)$ in Eq. (21). In particular, we compare the results of our default regularized training against the scenario when $\rho$, the coefficient of Reg$(R, I)$, is set to zero. Table 5 shows the results in terms of MAE. We observe that the inclusion of regularization, i.e., the presence of constraints, benefits the yield prediction.

| Method | NS1 | NS2 | NS3 | SC |
|---|---|---|---|---|
| $\rho = 0$ | 9.929 | 8.590 | 8.035 | 8.810 |
| YIELDNET | 9.245 | 8.387 | 7.914 | 8.751 |

Table 5: Effect of ablation of regularizer Reg$(R, I)$ in Eq. (21) (MAE).

## 6. Conclusion

In this work, we present YIELDNET, a novel model for yield prediction of the final product in multi-step reactions. The energy of the transition state plays a critical role in determining reaction yield. The graph structure of the transition state can be inferred using atom mapping between the reactants and products. As most datasets lack atom mapping, we employ a differentiable node alignment network, which can be trained end-to-end using supervision from yield rather than fine-grained atom mapping labels.

Our work opens several promising directions for future research. YIELDNET can be used to predict free energy, reaction rates, and more. Yield depends on factors beyond activation energy, like reaction time, purification procedure, etc., which current datasets lack. While the datasets also lack the underlying atom mapping, the graph structures of the reactants and products allowed us to approximate it. A key direction for future work is to incorporate the other factors into our model when such datasets become available.

## Impact Statement

This paper offers a valuable tool optimizing reaction yields. It can be used in prediction of yield when the datasets contain molecular graph structures. However, it is imperative to acknowledge the possibility of incorrect predictions by any yield prediction model, which demands human interventions. Additionally, misuse of the model may lead to predictions for undesirable compounds, potentially causing adverse societal consequences. Careful attention to ethical guidelines is paramount to ensure a responsible and positive impact on the model.

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

# Learning Condensed Graph via Differentiable Atom Mapping for Reaction Yield Prediction (Appendix)

## A. Limitations

1. Currently our work is limited to reactions having a reasonable number of atoms ($\sim 200 - 250$). However, implementing our model on larger molecular reactions or catalysis (Sheldon et al., 2020) has some drawbacks. Computation of $\boldsymbol{P}$ using Sinkhorn-iterations (5) has a complexity of $\mathcal{O}(N^2)$, where $N$ is the number of total atoms of reactant(s) or product(s) in a reaction step. Thus, computing $\boldsymbol{P}$ in such cases would be challenging. However, this issue can be mitigated by low-rank factorization (Scetbon et al., 2021) for Sinkhorn routine, which reduces the complexity to $\mathcal{O}(N)$.

2. The standard yield prediction datasets currently have only a single reaction path $r$ for reactions leads to one single product. Thus, we consider only single reaction path-based reactions to train our model. However, our method can be adapted to take multiple reactions into account, where we can add more stochasticity into Sinkhorn layers to enable generation of multiple permutations from the same set of reactants and products. They will lead to multiple atom mapping and CGRs, which in turn can generate multiple reaction paths.

3. The standard yield prediction datasets contain only molecular graphs of any molecular component and these datasets are commonly used in previous baselines. Hence, our model, as of now, does not incorporate information about geometry of the molecules. However, our model can be easily adapted to incorporate geometry by extending our GNN into the spatial graphs and taking into account three-dimensional coordinates for our atom mapping approximators.

## B. Additional discussion on related work

Apart from the general applications in chemistry mentioned in Section 2, there has been a recent interest in modeling chemical reactions (Bradshaw et al., 2019; Benayad et al., 2024; Joung et al., 2024; Hoque et al., 2024a) and solving reaction-related problems (Schwaller et al., 2021b; Burés & Larrosa, 2023). These also include retrosynthetic planning (Segler & Waller, 2017; Liu et al., 2017; Coley et al., 2017b; Dai et al., 2019; Shi et al., 2020; Chen et al., 2020; Yan et al., 2020; Fortunato et al., 2020; Somnath et al., 2021; Tu & Coley, 2022; Chen et al., 2023), reaction outcome prediction (Wei et al., 2016; Coley et al., 2017a; Jin et al., 2017; Schwaller et al., 2018; 2019; Coley et al., 2019; Struble et al., 2020; Guan et al., 2021; Bi et al., 2021; Chen & Jung, 2022; Nippa et al., 2023; 2024; King-Smith et al., 2024; Pereira et al., 2024), reaction optimization (Raccuglia et al., 2016; Zhou et al., 2017; Gao et al., 2018; Shields et al., 2021; Wang et al., 2024; Li et al., 2024; Hoque et al., 2024b), atom mapping (Schwaller et al., 2021a; Nugmanov et al., 2022; Chen et al., 2024; Astero & Rousu, 2024), transition state geometry prediction (Pattanaik et al., 2020; Lemm et al., 2021; Makoś et al., 2021; Choi, 2023; Duan et al., 2023; Kim et al., 2024; Duan et al., 2025), etc (Qian et al., 2023; M. Bran et al., 2024). In the case of atom mapping, RXNMapper (Schwaller et al., 2021a), and GraphormerMapper (Nugmanov et al., 2022) are the two pioneers of deep learning-based atom-mapping tools. RXNMapper uses post-processing on the attention heads, learned through an unsupervised task of reaction SMILES reconstruction. Due to this unsupervised nature, RXNMapper needs a substantial amount of data and a large transformer encoder-based model (Lan et al.). On the other hand, GraphormerMapper is pre-trained in multiple levels on different tasks, making the model very computationally expensive. As both models use attention-guided paths for atom mapping, these mappings aren't injective *i.e.*, there are multiple atom mapping possible for one single reaction instance and that should not happen in real case scenarios. Recently, Chen et al. (2024) use active learning for the atom mapping task, which is also attention-based. In another recent work, AMNet (Astero & Rousu, 2024) makes the atom mapping injective by imposing an extra layer of WL test algorithm (Weisfeiler & Leman, 1968) on top of a non-injective graph attention. However, YIELDNET model – (1) tries to approximate atom mapping as a doubly-stochastic matrix, which by architecture yields injective mapping, (2) is relatively simple. Our key goal is to predict yield which is also different from the above described atom mapping models. We achieve this by designing an architecture, that approximates atom-mapping on the fly.

# C. Neural architectures different components of YIELDNET

## C.1. Architecture of $\text{GNN}_\theta$

Our $\text{GNN}_\theta$ is built upon a communicative message passing network (CMPNN) (Song et al., 2020). In contrast to a simple message-passing neural network (MPNN) (Gilmer et al., 2017) or directed message-passing neural network (DMPNN) (Yang et al., 2019), where either only node states or edge states are updated independently in each iteration, our model updates both node and edge states simultaneously. Moreover, the update states for nodes and edges are treated as interdependent instead of treated independently. We elaborate on each step, starting from initial features as input to the final embedding as output.

In the following, we describe the embedding computation only for the reactant set $R$, which can subsequently be extended for the product set $I$. $\text{GNN}_\theta$ takes input graph $G_R$ as $\text{Adj}_R \in \{0,1\}^{N \times N}$, node features $\boldsymbol{V}_R \in \mathbb{R}^{N \times d_V}$, and edge features $\boldsymbol{E}_R \in \mathbb{R}^{N \times N \times d_E}$. We use two different MLPs for each of the features to transform into initial embeddings, $\text{nodeEmb}_0(u) \in \mathbb{R}^D$ and $\text{edgeEmb}_0(u,v) \in \mathbb{R}^D$ accordingly,

$$\text{nodeEmb}_0(u) = \text{MLP}_{\theta_1}(\boldsymbol{V}_R)[u] \tag{22}$$

$$\text{edgeEmb}_0(u,v) = \text{MLP}_{\theta_2}(\text{Adj}_R \odot \boldsymbol{E}_R)[u,v] \tag{23}$$

Here, $\text{MLP}_{\theta_.}$ are just single-layer networks followed by a ReLU activation function. In Eq. (23), $\text{Adj}_R \in \{0,1\}^{N \times N}$ and $\boldsymbol{E}_R \in \mathbb{R}^{N \times N \times d_E}$. Here, in $\odot$ operation, we first broadcast $\text{Adj}_R$ into the third dimension to have an adjacency tensor of dimension $N \times N \times d_E$ and then perform Hadamard product with $\boldsymbol{E}_R$. Next, we perform message passing for $k$ number of propagation layers. For a $k \in \{1,..,K-1\}$, the message passing steps are given by,

$$\boldsymbol{\mu}_k(v) = \sigma(\max_{u \in \text{nbr}(v)} \text{edgeEmb}_{k-1}(u,v)) \odot \sum_{u \in \text{nbr}(v)} \text{edgeEmb}_{k-1}(u,v) \tag{24}$$

$$\text{nodeEmb}_k(v) = \text{nodeEmb}_{k-1}(v) + \boldsymbol{\mu}_k(v); \quad \text{nodeEmb}_k(v) \in \mathbb{R}^D \tag{25}$$

$$\boldsymbol{\nu}_k(u,v) = \text{Adj}_R[u,v] \cdot \Big(\text{nodeEmb}_k(v) - \text{edgeEmb}_{k-1}(v,u)\Big) \tag{26}$$

$$\text{edgeEmb}_k(u,v) = \beta_k(u,v) \cdot \text{ReLU}\Big(\text{edgeEmb}_0(u,v) + \theta_{3,k} \cdot \boldsymbol{\nu}_k(u,v)\Big) \tag{27}$$

$$\text{where } \beta_k(u,v) = \frac{\exp\Big(\text{Adj}_R[u,v] \cdot \text{nodeEmb}_k(u)^\top \text{nodeEmb}_k(v)\Big)}{\sum_{u'} \exp\Big(\text{Adj}_R[u',v] \cdot \text{nodeEmb}_k(u')^\top \text{nodeEmb}_k(v)\Big)} \tag{28}$$

The above computations were performed for $k \in \{1,..K-1\}$. For $k = K$, we update the embeddings as follows:

$$\boldsymbol{\mu}_K(v) = \sigma(\max_{u \in \text{nbr}(v)} \text{edgeEmb}_{K-1}(u,v)) \cdot \sum_{u \in \text{nbr}(v)} \text{edgeEmb}_{K-1}(u,v) \tag{29}$$

$$\text{nodeEmb}_K(v) = \text{MLP}_{\theta_4}\Big(\boldsymbol{h}_0(v), \text{nodeEmb}_{K-1}(v), \boldsymbol{\mu}_K(v)\Big); \quad \text{nodeEmb}_K(v) \in \mathbb{R}^d \tag{30}$$

$$\text{edgeEmb}_K(u,v) = \text{edgeEmb}_{K-1}(u,v); \quad \text{edgeEmb}_K(v) \in \mathbb{R}^D \tag{31}$$

Finally, we compute node embedding vector $\boldsymbol{h}_R(u)$ per node $u$, node embedding matrix $\boldsymbol{H}_R \in \mathbb{R}^{N \times d}$, edge embedding vector $\boldsymbol{m}_R(u,v)$ for edge $(u,v)$, edge embedding matrix $\boldsymbol{M}_R \in \mathbb{R}^{N \times N \times D}$, as follows:

$$\boldsymbol{h}_R(u) = \text{nodeEmb}_K(u) \tag{32}$$

$$\boldsymbol{m}_R(u,v) = \text{edgeEmb}_K(u,v) \tag{33}$$

$$\boldsymbol{H}_R[u,:] := \boldsymbol{h}_R(u) \tag{34}$$

$$\boldsymbol{M}_R[u,v,:] := \boldsymbol{m}_R(u,v) \tag{35}$$

## C.2. Architecture of $\text{InputDifferentiableGNN}_\psi$

This architecture is similar to the $\text{GNN}_\theta$, except that we realize this in tensor space and use adjacency tensor $\text{Adj}_{\text{CGR}} \in \mathbb{R}^{N \times N \times D}$. This is because during the continuous approximation of $\text{Adj}_{\text{CGR}}$ in Eq. (12), we need $\boldsymbol{M}_R, \boldsymbol{M}_I \in \mathbb{R}^{N \times N \times D}$ (3). Thus, to obtain $\text{Adj}_{\text{CGR}}$, we need to do $\odot$ operation *i.e.*, broadcasting $\text{Adj}_\bullet \in \mathbb{R}^{N \times N}$ into a third dimension to have an adjacency tensor of $N \times N \times D$ followed by Hadamard multiplication with corresponding $\boldsymbol{M}_\bullet$. Finally, we obtain an $\text{Adj}_{\text{CGR}} \in \mathbb{R}^{N \times N \times D}$ through Eq. (12). Now, similar to the $\text{GNN}_\theta$, to perform the Eq. (23), we feed $\text{Adj}_{\text{CGR}}$ into an MLP

to bring its final dimension from $D$ to $d_E$ same as $\boldsymbol{E}$, so that the Hadamard product in Eq. (23) is compatible.

$$\widetilde{\mathrm{Adj}}_{\mathrm{CGR}} = \mathrm{MLP}_{\psi_0}(\mathrm{Adj}_{\mathrm{CGR}}) \tag{36}$$

$$\mathrm{nodeEmbT}_0 = \mathrm{MLP}_{\psi_1}(\boldsymbol{V}_R) \tag{37}$$

$$\mathrm{edgeEmbT}_0 = \mathrm{MLP}_{\psi_2}(\widetilde{\mathrm{Adj}}_{\mathrm{CGR}} \odot \boldsymbol{E}_{\mathrm{CGR}}) \tag{38}$$

Then the tensorized version for Eqs. (24)-(28) for $k \in \{1, ..., K-1\}$ is written as ($x\mathrm{T}$ indicates tensorized version of a vector $\boldsymbol{x}$ described in $\mathrm{GNN}_\theta$):

$$\mu\mathrm{T}_k[v,:] = \sigma\left(\max_u \mathrm{edgeEmbT}_{k-1}[u,v,:]\right) \odot \sum_u \mathrm{edgeEmbT}_{k-1}[u,v,:] \tag{39}$$

$$\mathrm{nodeEmbT}_k[v,:] = \mathrm{nodeEmbT}_{k-1}[v,:] + \mu\mathrm{T}_k[v,:] \tag{40}$$

$$\nu\mathrm{T}_k[u,v,:] = \mathrm{Adj}_{\mathrm{CGR}}[u,v,:] \odot \mathrm{nodeEmbT}_k[v,:] - \mathrm{edgeEmbT}_{k-1}^\top[u,v,:] \tag{41}$$

$$\mathrm{edgeEmbT}_k[u,v,:] = \beta_k[u,v,:] \odot \mathrm{ReLU}\left(\mathrm{edgeEmb}_0[u,v,:] + \psi_{3,k} \cdot \nu\mathrm{T}_k[u,v,:]\right) \tag{42}$$

$$\text{where } \beta\mathrm{T}_k[u,v,:] = \frac{\exp\left((\mathrm{nodeEmbT}_k \, \mathrm{nodeEmbT}_k^\top) \odot \mathrm{Adj}_{\mathrm{CGR}}\right)[u,v,:]}{\sum_{u'} \exp\left((\mathrm{nodeEmbT}_k \, \mathrm{nodeEmbT}_k^\top) \odot \mathrm{Adj}_{\mathrm{CGR}}\right)[u',v,:]} \tag{43}$$

For $k = K$, we update the embeddings as:

$$\mu\mathrm{T}_K[v,:] = \sigma(\max_u \mathrm{edgeEmbT}_{K-1}[u,v,:]) \odot \sum_u \mathrm{edgeEmbT}_{K-1}[u,v,:] \tag{44}$$

$$\mathrm{nodeEmbT}_K[v,:] = \mathrm{MLP}_{\psi_4}\left(\mathrm{nodeEmbT}_0[v,:], \mathrm{nodeEmbT}_{K-1}[v,:], \mu\mathrm{T}_K[v,:]\right) \tag{45}$$

Finally, $\boldsymbol{H}_{\mathrm{CGR}}$ is computed as $\mathrm{nodeEmbT}_K \in \mathbb{R}^{N \times d_H}$.

## C.3. Details about neural path encoder

Neural path encoder is composed of four components $\mathrm{Transformer}_{\phi_1}$, $\mathrm{NodeAggr}_{\phi_2}$, $\mathrm{StepAggr}_{\phi_3}$, and $\mathrm{MLP}_{\phi_4}$.

1. $\mathrm{Transformer}_{\phi_1}$ Before going to $\mathrm{Transformer}_{\phi_1}$ encoder, $\boldsymbol{H}_{\mathrm{CGR}_i}$ supposed to have a concatenation with $\mathbf{step_i}$, which is similar to the concept of positional encoding (Vaswani et al., 2017). Here, $i \leq n$, where $n$ is the number of total steps in reaction path $r$. However, we used one-hot positional encoding, which leads us to $\mathbf{step_i} \in \{0,1\}^{N \times n}$. Thus, according to Eq. (16), $\overline{\boldsymbol{H}}_i \in \mathbb{R}^{N \times (d_H + n)}$. As our $\mathrm{Transformer}_{\phi_1}$ is a single transformer encoder layer having the same encoder layer architecture as Vaswani et al. (2017), our final $\boldsymbol{S}_i \in \mathbb{R}^{N \times (d_H + n)}$ (17).

2. $\mathrm{NodeAggr}_{\phi_2}$ The main architecture is the same as Seq2Seq for sets or known as Set2Set (Vinyals et al., 2016). This type of aggregation inflates the final output dimension of embeddings to twice the input dimension *i.e.* $2(d_H + n)$. Thus, we use a Linear layer to reduce back the output dimension to the same as the input dimension $(d_H + n)$. This breaks down the Eq. (18) into the following

$$\boldsymbol{s}_i = \mathrm{Linear}_{\phi_{2\mathrm{b}}}\left(\mathrm{Set2Set}_{\phi_{2a}}(\boldsymbol{S}_i[1,:], ..., \boldsymbol{S}_i[N,:])\right), \text{ for } i \leq n. \tag{46}$$

3. $\mathrm{StepAggr}_{\phi_3}$ This architecture is also similar to $\mathrm{NodeAggr}_{\phi_2}$. Thus, Eq. (19) can be rewritten as,

$$\boldsymbol{z}_r = \mathrm{Linear}_{\phi_{3\mathrm{b}}}\left(\mathrm{Set2Set}_{\phi_{3a}}(\boldsymbol{s}_1, ..., \boldsymbol{s}_n)\right); \quad \boldsymbol{z}_r \in \mathbb{R}^{(d_H + n)} \tag{47}$$

4. $\mathrm{MLP}_{\phi_4}$ The final $\mathrm{MLP}_{\phi_4}$ layer is a $\mathrm{Linear} - \mathrm{ReLU} - \mathrm{Linear}$ (LRL) layer, the input $\boldsymbol{z}_r$ has a final dimension of $(d_H + n)$.

# D. Additional details about experimental setup

## D.1. Dataset preparation

To assess the efficacy of our model, we focus on four reactions that have garnered recent attention. These reactions— Gas-phase isomerization (GP) (Grambow et al., 2020b), deoxyflurorination (DF) (Nielsen et al., 2018), asymmetric N,S-acetal formation (NS) (Zahrt et al., 2019), and Suzuki coupling reaction (SC)— exhibit diverse data sizes and varying number of steps.

Each reaction in these datasets is annotated with sets of reactants, intermediates, and products based on mechanistic considerations. The measured yields ranging from 0% to 100% are compiled from prior wet-lab experimental data, except in the case of the GP dataset, which we will discuss shortly. Several of these datasets, including (DF) (Nielsen et al., 2018) and asymmetric N, S-acetal formation (NS) (Zahrt et al., 2019), are obtained through High-Throughput Experimentation (HTE). The experimental process for HTE typically involves selecting a limited set of reactants and conducting reactions on *all possible combinations within those chosen reactants.* In contrast, real-life datasets contain a significantly larger pool of diverse reactants, with only a small fraction of the potential reactant combinations explored. This discrepancy results in a substantial reduction in dataset size for real-life scenarios (Saebi et al., 2021; Schleinitz et al., 2022). Additionally, real-life datasets exhibit higher sparsity, making it less likely to encounter a reactant from the test sample in the training set.

Due to the above issues, we create subsets of these HTE datasets, which create multiple low throughput (LTE) datasets, making them closer to reality. Specifically, we derive GP, GP1, GP2 from the gas-phase isomerization dataset; NS1, NS2, NS3, NS4, NS5 from the NS dataset; DF1, DF2 and DF3 from DF dataset. Moreover, we collect a new dataset consisting of samples of the reaction type Suzuki Coupling (SC) using the reaction data from $\sim 50$ peer reviewed publications. In contrast to the existing HTE datasets, this dataset is low throughput, *i.e.*, they consist of reactions of only a limited number of combinations between the reactants. We also use USPTO from Lowe (2017). These finally lead to total fifteen datasets.

### D.1.1. LIST OF DATASETS

The fifteen datasets used are: GP, GP1, GP2, USPTO, NS1, NS2, NS3, NS4, NS5, NS, DF1, DF2, DF3, DF, and SC. Among them, we present GP, NS1, NS2, NS3, NS4, NS5, SC and DF in the main part and present the rest in this Appendix. In the following part, we describe the datasets in full detail.

### D.1.2. DATASETS DERIVED FROM GAS PHASE ISOMERIZATION

The gas-phase isomerization reaction dataset (Grambow et al., 2020b) contains around 12000 elementary step organic reactions involving small molecules composed solely of C, H, N and O atoms. The dataset has the activation energy for the reaction as the label. Previous attempts to predict activation energies involved using fingerprints and graph-based models (Choi et al., 2018; Grambow et al., 2020a; Heid & Green, 2021). We transformed the activation energy ($\Delta \mathcal{E}^{\ddagger}$) into yield values, by using the Eyring equation (Eyring, 1935). This approximates a rate constant ($\gamma$) value from activation energy. We then assume every elementary step reaction following first-order kinetics to get the yield value. The steps are shown below. Notations used in Eqs. (48)— (51) are only limited to this section and may share overlap with symbols used in other parts of the paper. Here, $\Delta G^{\ddagger}$ represents Gibbs free energy of activation (or assumed to be activation energy) in Jmol$^{-1}$. $k_B, h, R$ are the Boltzmann constant ($1.38 \times 10^{-23}$ JK$^{-1}$), Planck constant ($6.626 \times 10^{-34}$Js), and universal gas constant ($8.314$ JKmol$^{-1}$) respectively. Here, $t$ here is the reaction time, we take a fix $t$ of 1 hour i.e., 3600s.

$$\gamma = \frac{k_B \text{Temp}}{h} \exp\left(\frac{-\Delta G^{\ddagger}}{R \cdot \text{Temp}}\right) \tag{48}$$

$$\text{yield} = (1 - \exp(-\gamma t)) \times 100 \tag{49}$$

Temp is the reaction temperature. In our case, we consider a fixed Temp $= 1105.26$K for our GP dataset. We have activation energy in this dataset in terms of kcal mol$^{-1}$. To convert it to a proper unit, we multiply our activation energy value by a factor of 4183 and feed the values in place of $\Delta G^{\ddagger}$. After all these, our final equation becomes,

$$\gamma = \frac{1.38 \times 10^{-23} \times 1105.26}{6.626 \times 10^{-34}} \exp\left(\frac{-\Delta \mathcal{E}^{\ddagger} \times 4183}{8.314 \times 1105.26}\right) \tag{50}$$

$$\text{yield} = (1 - \exp(-\gamma \times 3600)) \times 100 \tag{51}$$

We derive GP and its other two variants GP1 and GP2 based on clustering of the gas-phase reaction dataset. We observe that the reactants in GP1 mostly contain very small, 3 or 4-membered rings (e.g., cyclopropanes, oxiranes, aziridines,

cyclobutanes, etc.). Similarly, reactants in GP2 are mostly substituted *five-membered aromatic heterocycles* (e.g., pyrrole, pyrazole, imidazole, furan, etc.). In total, we have used three datasets GP (main), GP1 (Appendix), and GP2 (Appendix).

### D.1.3. DATASETS DERIVED FROM DEOXYFLUORINATION

The deoxyfluorination (DF) reaction is an important method for converting inexpensive alcohol to the corresponding fluorinated compounds by using sulfonyl fluorides, in the presence of a suitable base (Nielsen et al., 2018). Here, the whole dataset represents a collection of combination reactions involving 37 unique alcohols, 5 sulfonyl fluoride, and 4 bases, totaling 740 reactions and their measured yields. Previous studies on yield prediction for this reaction employed the random forest (Nielsen et al., 2018) and NLP-based transfer learning (Singh & Sunoj, 2022). Apart from the full dataset (DF), we have extracted three subsets, each containing 200 instances, from the full dataset. We create these subsets based on the yield labels. Yields have a skewed distribution, where yield has a mean around 40. We create three subsets having an average yield of low (10), medium (75) and high (90). We find that there are different predominant compounds. For instance:

(1) The dataset DF1 has 2-(4-hydroxycyclohexyl)isoindoline-1,3-dione, 2,3,4,6,7,8,9,10-octahydropyrimido[1,2-a]azepine, 4-nitro benzenesulfonyl fluoride as predominant reactants.

(2) The dataset DF2 has 3-([1,2,4]triazolo[1,5-a]pyrimidin-6-yl)propan-1-ol, 2-(tert-butyl)-1,1,3,3- tetramethylguanidine, 1,1,2,2,3,3,4,4,4-nonafluorobutane -1-sulfonyl fluoride predominant reactants.

(3) The dataset DF3 has 3-(4,5-diphenyloxazol-2-yl)propan-1-ol, 2-(tert-butyl)-1,1,3,3- tetramethylguanidine, 1,1,2,2,3,3,4,4,4-nonafluorobutane -1-sulfonyl fluoride as predominant reactants.

In total, we have used four datasets DF (main), DF1 (Appendix), DF2 (Appendix), and DF3 (Appendix).

### D.1.4. DATASETS DERIVED FROM ASYMMETRIC N, S-ACETAL FORMATION

The asymmetric N, S-acetal formation reaction (NS) is a catalytic transformation involving the addition of a thiol to aldimines. Zahrt et al. (2019) released a dataset comprising 1075 such examples, where the labels are in terms of activation barrier difference. Later, Singh & Sunoj (2022) released another version of the prior dataset, which comprises 1027 samples having labels as percentage values. This dataset consists of 5 thiols, 15 different imines as substrates, and 43 different chiral phosphoric acids as catalysts (Cat). The label for this dataset is expressed in terms of enantioselectivity, which is effectively the yield of the major enantiomer produced in the reaction. Similarly to the previous dataset, we have sampled the complete dataset into five subsets of size 300. Like DF, we create datasets with average yield of 25, 40, 50, 60, 75.

(1) The dataset NS1 has 2,6-dibromo-4-hydroxy-8,9,10,11,12,13,14,15-octahydrodinaphtho[2,1-d:1',2'-f][1,3,2] dioxaphosphepine-4-oxide; (E)-N-(2,4-dichlorobenzylidene)benzamide; ethanethiol as predominant reactants.

(2) The dataset NS2 has 2,6bis(anthracen-9-ylmethyl)-4hydroxydinaphtho[2,1-d:1',2'-f][1,3,2] dioxaphosphepine-4-oxide; (E)-N-benzylidenebenzamide; cyclohexanethiol as predominant reactants.

(3) The dataset NS3 has 2,6-bis(anthracen-9-ylmethyl)-4-hydroxydinaphtho[2,1-d:1',2'-f][1,3,2] dioxaphosphepine-4-oxide; (E)-N-(naphthalen-1-ylmethylene)benzamide; cyclohexanethiol as predominant reactants.

(4) The dataset NS4 has 4-hydroxy-2,6-bis(2-(naphthalen-2-yl)phenyl)-8,9,10,11,12,13,14,15-octahydrodinaphtho [2,1-d:1',2'-f][1,3,2]dioxaphosphepine-4-oxide; (E)-N-(4-methoxybenzylidene)benzamide; cyclohexanethiol as predominant reactants.

(5) The dataset NS5 has 4-hydroxy-2,6-bis(4-methoxyphenyl) dinaphtho[2,1-d:1',2'-f][1,3,2] dioxaphosphepine-4-oxide; (E)-N-(4-methoxybenzylidene)benzamide; 2-methylbenzenethiol} as predominant reactants.

We also utilized the full dataset (abbreviated as NS) to check the performance. In total, we have six datasets NS1 (main), NS2 (main), NS3 (main), NS4 (main), and NS5 (main), and NS (Appendix).

### D.1.5. DATASETS DERIVED FROM SUZUKI COUPLING REACTION

The palladium-catalyzed Suzuki Cross-coupling (SC) is a versatile method for generating biaryl products—ubiquitous in natural compounds, pharmaceuticals, and chiral ligands (Kantchev et al., 2007). This dataset comprising 481 reactions is manually curated from 25 publications filtered out from numerous reports within this reaction class (Çakır et al., 2021; O'Brien et al., 2006; Navarro et al., 2006; Li et al., 2019a; Peh et al., 2010; Chen & Kao, 2017; Lu et al., 2017; Micksch

et al., 2014; Diebolt et al., 2010; Tu et al., 2012; Ouyang et al., 2018; Han et al., 2018; Zhang et al., 2020; Karataş, 2019; Yang, 2017; Lv et al., 2014; Organ et al., 2009; Xia et al., 2021; Navarro et al., 2004; Wu et al., 2011; Sahin et al., 2017; Kuriyama et al., 2013; Hartmann et al., 2009; Dastgir et al., 2010; Arıcı et al., 2021; Azpiroz et al., 2017; Kumar et al., 2009; Mercan et al., 2011; Izquierdo et al., 2015). The selection criteria were specifically tailored to a type of catalyst where palladium is intricately bound to an N-heterocyclic carbene (NHC) ligand, with additional selection criteria involving heteroatoms (N, O, and P) within the catalyst structure. The dataset is labeled based on the yield of the resulting biaryl product. We used the SC dataset in our main paper.

### D.1.6. DATASETS DERIVED FROM USPTO

This is a US patent chemical reaction dataset collected by Lowe (2017). Later the dataset is filtered to classify the reaction classes (Schwaller et al., 2021b). The reaction dataset has around 44k reactions with yield labels and reaction classes. We use these reaction classes to filter out the dataset. With the further exclusion of reactions with missing reagents, we selected 1700 reaction samples from the original dataset. Next, we select only those reactions which occur through a two-step reaction pathway. For instance, the protection of functional groups (e.g., alcohol, amines), functional group interconversion, including nucleophilic substitution, etc. Finally, we curated 1150 two-step reaction samples for our study. We used the USPTO dataset in the Appendix.

### D.2. Details about our implementations of the baselines

While implementing the baselines, we performed few modifications to ensure that the comparison between our method and the baselines is fair— (1) number of parameters is approximately same; and (2) no component of a baseline is exposed/pre-trained in external dataset, since that would give additional signals to them, which is not provided to our method and rest of the baselines.

**GCN**   Here in GNN, the message function employs the graph convolution layer (GCN) (Gilmer et al., 2017), which collects adjacency node embeddings through edges. We have developed other versions of it by replacing this GCN with alternative node-based convolution message functions. Finally, following Kwon et al. (2022), we predict the yield by taking molecular graphs of reactants and products as input. It aggregates reactant vectors, concatenates them with the product vector, and channels them through an FNN to predict the yield. We set the dimension of node embeddings as 18.

**HGT**   Here in GNN, we utilize a Heterogeneous Graph Transformer-based Convolution (HGT) as our message function to treat the molecular graph as a heterogeneous graph (Hu et al., 2020). Then, we used the same method proposed by Kwon et al. (2022), which is described in the context of GCN. We set the dimension of node embeddings as 17.

**TAG**   Here in GNN, we utilize TAG, which is based on the topology adaptive graph convolution network (Du et al., 2017). Then, we used the same method proposed by Kwon et al. (2022), which is described in the context of GCN. We set the dimension of node embeddings as 20.

**GIN**   Here in GNN, we employ the GIN message convolution function, inspired by graph isomorphism networks, showcasing a Graph Neural Network (GNN) that exhibits comparable power to the Weisfeiler-Lehman (WL) test for isomorphism (Xu et al.). Then, we used the same method proposed by Kwon et al. (2022), which is described in the context of GCN. We set the dimension of node embeddings as 20.

**DeepReac+**   The overall architecture of the (Gong et al., 2021) model is founded on a two-level graph attention convolution framework (Veličković et al., 2018). In the initial stage, the graphs for each reaction component are encoded using a graph attention network (GAT)-based GNN. Later, these individual component graphs are treated as nodes within a fully connected reaction graph, using the graph embeddings serving as node features. Another layer of GAT is then applied to this complete graph. Finally, the embeddings of all components are fed into a FFN network. We set the dimension of node embeddings as 48.

**YieldBERT**   YieldBERT uses a previously trained BERT encoder (Devlin et al., 2019) to forecast chemical reaction yield as a function of reaction SMILES (Weininger, 1988). But that would expose the model to external datasets possibly containing examples in test data too. Hence, we train BERT encoder along with a regressor, end-to-end for yield prediction. We set the dimension of hidden size as 12.

## D.3. Implementation details

**Training**  We maintain a 70:10:20 split for training, validation, and testing across all models, employing 10 different splits for robust evaluation. We kept the batch size $b$ same across all models. We set $b = 50$ for GP datasets and $b = 8$ for the rest. We train each model with 100 epochs and evaluate their performance on the test dataset, selecting the epoch with the lowest validation MAE. We use Adam optimizer for each model. We use Noam learning rate with 2 warmup epochs and an initial and final learning rate of $10^{-4}$ and a maximum learning rate of $10^{-3}$. Additionally, we keep the regularizer parameter $\rho$ fixed at a value of 0.1 in our model.

**Model**

1. $\text{GNN}_\theta$. We set the dimension of node embeddings, $d = 20$ (30) and dimension of edge embeddings, $D = 20$ (31).
2. Align. We set temperature $\lambda = 0.1$ in Eq. (6), Gumbel noise factor 1.0, and Sinkhorn iterations $T = 10$.
3. $\text{InputDifferentiableGNN}_\psi$. It shares parameters with $\text{GNN}_\theta$ except $\psi_0$ and $\psi_4$ (Appendix C). We set $d_H = 20 - n$, where $n$ is the number of steps.
4. $\text{Transformer}_{\phi_1}$. As $d_H = 20 - n \implies (d_H + n) = 20$, which is the input and output dimension of the $\text{Transformer}_{\phi_1}$. We set the number of heads to 5, and the feedforward dimension to 2048.
5. $\text{NodeAggr}_{\phi_2}$ and $\text{StepAggr}_{\phi_3}$. As described in the Appendix C, the set encoders of Eqs. (46) and (47) of both the aggregator modules are borrowed off-the-shelf from Vinyals et al. (2016), where we have input dimension $(d_H + n) = 20$, number of layers 1, number of iterations 3. The final output of the Aggr modules will be of the dimension of $(d_H + n) = 20$.
6. $\text{MLP}_{\phi_4}$. We have input dimension of $(d_H + n) = 20$ and output dimension of 1 as per Eq. (20).

**Baseline**  For a fair comparison, we maintain a similar number of parameters across all the models. We used the same learning rate for all GNN models and YieldBERT of $10^{-4}$. For DeepReac+, the learning rate is $10^{-3}$.

**Number of parameters**  The number of parameters for multi-step reactions is around 112k, and almost 26k for a single step one. The number of parameters is almost maintained throughout all the models (Table 6).

| #Parameters | GP | rest |
|---|---|---|
| GCN | 26825 | 115753 |
| HGT | 28405 | 151895 |
| TAG | 27965 | 108725 |
| GIN | 27165 | 105525 |
| DeepReac+ | 27074 | 119522 |
| YieldBERT | 28861 | 110797 |
| YIELDNET | 26478 | 112583 |

Table 6: Number of parameters for all models

## D.4. Hardware details

All the models are trained on NVIDIA A100 80GB GPU. All the models are fully based on PyTorch (Paszke et al., 2019). We run in Ubuntu 20.04.6 LTS machine having 2TB RAM with 64 bit CPU and AMD EPYC 7742 64-Core Processor.

## D.5. License

We utilize CMPNN (Song et al., 2020), which comes under MIT License. For baseline comparisons, we use dgl-based (Wang et al., 2019a) GCN (Kwon et al., 2022), HGT (Hu et al., 2020), TAG (Du et al., 2017), GIN (Xu et al.), and DeepReac+ (Gong et al., 2021) - all of which come under the Apache 2.0 License. YieldBERT (Schwaller et al., 2021b) comes under MIT License. The Gas-Phase reaction datasets (Grambow et al., 2020b) are licensed under CC-BY 4.0. The USPTO dataset used here comes under the MIT License. The original USPTO dataset (Schwaller et al., 2021b) by Lowe (2017) comes under CC0 1.0 License. RXNMapper (Schwaller et al., 2021a) tool comes under MIT License. RDKit tool comes under BSD-3-Clause License. NS (Zahrt et al., 2019) and DF (Nielsen et al., 2018) datasets used here, are available in (Singh & Sunoj, 2022). However, we couldn't find licenses for those datasets. PyTorch (Paszke et al., 2019) and NetworkX (Hagberg et al., 2008) both come under BSD-3-Clause License.

# E. Additional results

## E.1. Comparison in yield prediction in terms of RMSE

From Table 7 we observe a similar trend as Table 1 in RMSE for seven out of eight datasets. A different trend in RMSE for GP is due to using MAE on the validation set during early stopping. The metric used for early stopping was MAE in the validation set, which is why it may show an alternate trend for RMSE in GP. Like Table 1, YIELDNET (sky) (skyline variants of our model, which uses true CGR for yield prediction) outperforms other models, reflecting CGR's importance in yield prediction.

| Model | GP | NS1 | NS2 | NS3 | NS4 | NS5 | SC | DF |
|---|---|---|---|---|---|---|---|---|
| GCN | $42.778 \pm 0.276$ | $15.080 \pm 1.151$ | $12.090 \pm 1.011$ | $10.438 \pm 0.864^*$ | $11.667 \pm 0.286$ | $6.631 \pm 0.532$ | $16.314 \pm 0.762$ | $16.245 \pm 0.392$ |
| HGT | $43.442 \pm 0.181$ | $17.550 \pm 1.038$ | $12.241 \pm 1.012$ | $11.222 \pm 0.758$ | $11.639 \pm 0.334$ | $6.805 \pm 0.552$ | $19.136 \pm 0.612$ | $23.358 \pm 0.195$ |
| TAG | $41.698 \pm 0.336$ | $17.470 \pm 0.972$ | $12.212 \pm 0.993$ | $10.953 \pm 0.798$ | $11.575 \pm 0.318$ | $6.766 \pm 0.563$ | $18.792 \pm 0.661$ | $22.177 \pm 0.473$ |
| GIN | $42.712 \pm 0.351$ | $17.449 \pm 1.052$ | $12.285 \pm 1.011$ | $10.760 \pm 0.799$ | $11.424 \pm 0.290$ | $6.782 \pm 0.552$ | $18.467 \pm 0.635$ | $21.320 \pm 0.428$ |
| DeepReac+ | $35.748 \pm 0.375$ | $16.349 \pm 1.474$ | $13.048 \pm 0.699$ | $12.041 \pm 0.960$ | $12.526 \pm 0.921$ | $6.992 \pm 0.391^*$ | $19.852 \pm 1.604$ | $17.188 \pm 1.680$ |
| YieldBERT | $44.733 \pm 0.260$ | $16.438 \pm 1.153$ | $12.090 \pm 0.992$ | $10.955 \pm 0.894^*$ | $11.971 \pm 0.342$ | $6.803 \pm 0.557$ | $16.104 \pm 0.620$ | $15.993 \pm 0.384$ |
| YIELDNET | $38.195 \pm 0.436$ | $12.415 \pm 0.866$ | $10.701 \pm 0.966$ | $9.702 \pm 0.969$ | $9.383 \pm 0.491$ | $6.507 \pm 0.535$ | $14.029 \pm 0.795$ | $9.231 \pm 0.258$ |
| YIELDNET(sky) | $31.598 \pm 0.448$ | NA | NA | NA | NA | NA | NA | NA |

Table 7: Comparison of yield prediction performance for YIELDNET against all the competitive baselines, *viz.*, GCN (Kipf & Welling, 2017), HGT (Hu et al., 2020), TAG (Du et al., 2017), GIN (Xu et al.), DeepReac+ (Gong et al., 2021), YieldBERT (Schwaller et al., 2021c), on the 20% test examples, across all datasets. Performance is measured in terms of Root Mean Squared Error (RMSE). Numbers in green (yellow) indicate the best (second best) performer. Our improvement in performance over the next best baseline, where YIELDNET is the best performer, is statistically significant with p-value $< 0.05$, except in the cases marked with $^*$.

## E.2. Statistical significance test

To check the statistical significance of our results, we perform the paired t-test between YIELDNET and each baseline. We report the p-value of MAE and RMSEs for Table 1 and Table 7 in Table 8. If the p-value between the performance metrics is lower than the 5e-2 margin, we consider the corresponding performance significant.

| p-value for MAE table | | | | | | | | |
|---|---|---|---|---|---|---|---|---|
| Model | GP | NS1 | NS2 | NS3 | NS4 | NS5 | SC | DF |
| GCN | 1.5e-10 | 3.6e-03 | 7.7e-03 | 2.7e-02 | 1.6e-02 | 3.7e-01* | 1.6e-05 | 1.1e-07 |
| HGT | 5.1e-11 | 3.1e-05 | 1.7e-03 | 2.3e-03 | 2.5e-02 | 3.7e-02 | 1.0e-07 | 5.5e-12 |
| TAG | 2.1e-09 | 1.7e-05 | 1.6e-03 | 4.6e-03 | 2.7e-02 | 3.9e-02 | 3.3e-06 | 4.2e-10 |
| GIN | 1.0e-09 | 2.8e-05 | 1.8e-03 | 5.8e-03 | 2.0e-02 | 1.1e-02 | 7.8e-07 | 4.8e-09 |
| DeepReac+ | 3.3e-06 | 3.7e-03 | 2.4e-03 | 6.5e-02* | 1.8e-02 | 9.3e-02* | 1.0e-03 | 6.5e-03 |
| YieldBERT | 9.6e-11 | 1.8e-04 | 4.3e-04 | 9.8e-02* | 1.5e-02 | 4.9e-02 | 1.0e-03 | 3.9e-07 |

| p-value for RMSE table | | | | | | | | |
|---|---|---|---|---|---|---|---|---|
| Model | GP | NS1 | NS2 | NS3 | NS4 | NS5 | SC | DF |
| GCN | 3.7e-06 | 4.1e-03 | 3.5e-03 | 9.2e-02* | 4.8e-03 | 3.1e-02 | 3.0e-04 | 8.2e-08 |
| HGT | 1.6e-06 | 9.8e-06 | 1.9e-03 | 6.0e-03 | 5.5e-03 | 8.3e-03 | 4.1e-06 | 4.6e-12 |
| TAG | 1.5e-04 | 8.7e-06 | 1.9e-03 | 1.5e-02 | 5.8e-03 | 2.7e-02 | 1.1e-05 | 2.8e-10 |
| GIN | 6.7e-06 | 1.3e-05 | 1.7e-03 | 2.5e-02 | 7.9e-03 | 2.5e-02 | 6.9e-06 | 6.5e-10 |
| DeepReac+ | 3.2e-02 | 4.9e-03 | 3.1e-03 | 2.6e-02 | 7.9e-03 | 1.7e-01* | 2.1e-02 | 4.6e-03 |
| YieldBERT | 3.0e-07 | 1.9e-04 | 1.0e-03 | 5.2e-02* | 6.5e-03 | 3.1e-02 | 2.0e-03 | 7.7e-08 |

Table 8: p-value for MAE and RMSEs for all baseline models compared to our model. Our improvement in performance over the next best baseline, where YIELDNET is the best performer, is statistically significant with p-value $< 0.05$, except the cases marked with $^*$.

### E.3. Additional results on comparison across various atom-to-atom alignment methods

Here, we report the MAE and RMSE with standard error for all atom-to-atom alignment methods introduced in Section 5.2. Table 9 and 10 report the performances, which reveal that our alignment method is the best performer in the majority of the cases. Due to diversity in the reaction for GP and SC, they don't follow a common reaction template, thus atom-mapping using RDKit aren't feasible for them.

| Mean Absolute Error (MAE) | | | | |
|---|---|---|---|---|
| **Model** | **GP** | **NS1** | **NS2** | **NS3** |
| RDKit | – | $9.910 \pm 0.630$ | $8.845 \pm 0.873$ | $8.567 \pm 0.931$ |
| RXNMapper | $20.604 \pm 0.433$ | $9.610 \pm 0.602$ | $8.871 \pm 0.871$ | $8.470 \pm 0.970$ |
| Random | $25.187 \pm 0.407$ | $11.195 \pm 0.890$ | $8.984 \pm 0.993$ | $8.695 \pm 0.796$ |
| Attention | $25.730 \pm 0.758$ | $9.548 \pm 0.414$ | $9.024 \pm 0.894$ | $8.432 \pm 0.934$ |
| YIELDNET | $23.152 \pm 0.393$ | $9.245 \pm 0.518$ | $8.387 \pm 0.907$ | $7.914 \pm 0.931$ |

| Mean Absolute Error (MAE) | | | | |
|---|---|---|---|---|
| **Model** | **NS4** | **NS5** | **SC** | **DF** |
| RDKit | $7.237 \pm 0.375$ | $4.533 \pm 0.246$ | – | $7.187 \pm 0.208$ |
| RXNMapper | $6.871 \pm 0.406$ | $4.677 \pm 0.315$ | $10.046 \pm 0.434$ | $6.879 \pm 0.173$ |
| Random | $7.991 \pm 0.256$ | $4.504 \pm 0.243$ | $10.223 \pm 0.474$ | $10.595 \pm 0.244$ |
| Attention | $6.955 \pm 0.259$ | $4.317 \pm 0.220$ | $8.826 \pm 0.399$ | $7.593 \pm 0.220$ |
| YIELDNET | $7.015 \pm 0.495$ | $4.382 \pm 0.249$ | $8.751 \pm 0.438$ | $6.941 \pm 0.192$ |

Table 9: Comparison between different alignment strategies in terms of MAE of yield prediction. Brief results were presented in Table 2. Numbers in green (yellow) indicate the best (second best) performer.

| Root Mean Squared Error (RMSE) | | | | |
|---|---|---|---|---|
| **Model** | **GP** | **NS1** | **NS2** | **NS3** |
| RDKit | – | $13.133 \pm 0.802$ | $11.381 \pm 0.910$ | $10.451 \pm 0.912$ |
| RXNMapper | $35.429 \pm 0.455$ | $12.769 \pm 0.854$ | $11.366 \pm 0.852$ | $10.365 \pm 1.002$ |
| Random | $37.428 \pm 0.482$ | $14.827 \pm 1.185$ | $11.467 \pm 0.998$ | $10.852 \pm 0.772$ |
| Attention | $39.513 \pm 0.659$ | $12.703 \pm 0.713$ | $11.484 \pm 0.933$ | $10.500 \pm 0.933$ |
| YIELDNET | $38.195 \pm 0.436$ | $12.415 \pm 0.866$ | $10.701 \pm 0.966$ | $9.702 \pm 0.969$ |

| Root Mean Squared Error (RMSE) | | | | |
|---|---|---|---|---|
| **Model** | **NS4** | **NS5** | **SC** | **DF** |
| RDKit | $9.972 \pm 0.599$ | $6.732 \pm 0.557$ | – | $9.770 \pm 0.334$ |
| RXNMapper | $9.483 \pm 0.740$ | $6.984 \pm 0.684$ | $15.486 \pm 0.773$ | $9.202 \pm 0.221$ |
| Random | $10.908 \pm 0.421$ | $6.644 \pm 0.562$ | $15.419 \pm 0.839$ | $14.098 \pm 0.324$ |
| Attention | $9.531 \pm 0.421$ | $6.423 \pm 0.565$ | $13.719 \pm 0.710$ | $10.533 \pm 0.382$ |
| YIELDNET | $9.383 \pm 0.491$ | $6.507 \pm 0.535$ | $14.029 \pm 0.795$ | $9.231 \pm 0.258$ |

Table 10: Comparison between different alignment strategies in terms of RMSE of yield prediction. Brief results were presented in Table 2. Numbers in green (yellow) indicate the best (second best) performer.

### E.4. Additional results on ablation study on CGR representations

Here, we present the MAE and RMSE values for the ablation study on CGR computation introduced in Section 5.2, which extends the results of Table 3. From Table 11 we observe that the influence of CGR computation is benefitting on the yield prediction task for almost all of the datasets.

### E.5. Additional results on ablation study of reaction path encoder components

Next, we present the MAE and RMSE values for the ablation study on components of the reaction encoder for all the datasets (Section 5.2), which extends the results of Table 4. From Table 12 we can observe that the influence of the SetEncoders is way more significant than that of the Transformer. Since the GP is a single-step reaction dataset, these ablation studies are not applicable to it.

| | Mean Absolute Error (MAE) | | | | | | |
|---|---|---|---|---|---|---|---|
| **Method** | **GP** | **NS1** | **NS2** | **NS3** | **NS4** | **NS5** | **SC** | **DF** |
| w/o CGR | $24.921 \pm 0.427$ | $11.356 \pm 0.702$ | $9.459 \pm 0.953$ | $9.342 \pm 0.822$ | $7.726 \pm 0.428$ | $4.495 \pm 0.270$ | $10.686 \pm 0.453$ | $9.064 \pm 0.242$ |
| YIELDNET | $23.152 \pm 0.393$ | $9.245 \pm 0.518$ | $8.387 \pm 0.907$ | $7.914 \pm 0.931$ | $7.015 \pm 0.495$ | $4.382 \pm 0.249$ | $8.751 \pm 0.438$ | $6.941 \pm 0.192$ |

| | Root Mean Squared Error (RMSE) | | | | | | |
|---|---|---|---|---|---|---|---|
| **Method** | **GP** | **NS1** | **NS2** | **NS3** | **NS4** | **NS5** | **SC** | **DF** |
| w/o CGR | $36.686 \pm 0.353$ | $15.249 \pm 0.881$ | $12.075 \pm 0.949$ | $11.316 \pm 0.775$ | $10.506 \pm 0.54$ | $6.679 \pm 0.585$ | $16.134 \pm 0.748$ | $11.827 \pm 0.276$ |
| YIELDNET | $38.195 \pm 0.436$ | $12.415 \pm 0.866$ | $10.701 \pm 0.966$ | $9.702 \pm 0.969$ | $9.383 \pm 0.491$ | $6.507 \pm 0.535$ | $14.029 \pm 0.795$ | $9.231 \pm 0.258$ |

Table 11: Ablation study CGR computation. Brief results were introduced in Table 3. Numbers in green indicate the best performer.

| | Mean Absolute Error (MAE) | | | | | | |
|---|---|---|---|---|---|---|---|
| **Model** | **NS1** | **NS2** | **NS3** | **NS4** | **NS5** | **SC** | **DF** |
| StepAggr = DeepSet | $9.643 \pm 0.703$ | $9.008 \pm 0.915$ | $8.283 \pm 0.948$ | $6.124 \pm 0.202$ | $4.395 \pm 0.253$ | $8.551 \pm 0.292$ | $6.712 \pm 0.264$ |
| StepAggr = SumAggr | $10.192 \pm 0.743$ | $8.858 \pm 0.806$ | $8.349 \pm 0.909$ | $6.937 \pm 0.490$ | $4.433 \pm 0.262$ | $8.853 \pm 0.309$ | $7.165 \pm 0.183$ |
| Without transformer | $9.467 \pm 0.516$ | $8.680 \pm 0.786$ | $8.164 \pm 0.849$ | $6.407 \pm 0.224$ | $4.461 \pm 0.252$ | $8.628 \pm 0.504$ | $6.448 \pm 0.140$ |
| YIELDNET | $9.245 \pm 0.518$ | $8.387 \pm 0.907$ | $7.914 \pm 0.931$ | $7.015 \pm 0.495$ | $4.382 \pm 0.249$ | $8.751 \pm 0.438$ | $6.941 \pm 0.192$ |

| | Root Mean Squared Error (RMSE) | | | | | | |
|---|---|---|---|---|---|---|---|
| **Model** | **NS1** | **NS2** | **NS3** | **NS4** | **NS5** | **SC** | **DF** |
| StepAggr = DeepSet | $12.493 \pm 0.930$ | $11.594 \pm 0.912$ | $10.102 \pm 0.928$ | $8.297 \pm 0.250$ | $6.451 \pm 0.543$ | $13.117 \pm 0.584$ | $8.924 \pm 0.309$ |
| StepAggr = SumAggr | $13.341 \pm 0.998$ | $11.413 \pm 0.916$ | $10.078 \pm 0.910$ | $9.222 \pm 0.596$ | $6.62 \pm 0.615$ | $13.442 \pm 0.551$ | $9.531 \pm 0.223$ |
| Without transformer | $12.407 \pm 0.727$ | $10.871 \pm 0.845$ | $10.055 \pm 0.878$ | $8.900 \pm 0.424$ | $6.577 \pm 0.577$ | $13.485 \pm 0.777$ | $8.761 \pm 0.182$ |
| YIELDNET | $12.415 \pm 0.866$ | $10.701 \pm 0.966$ | $9.702 \pm 0.969$ | $9.383 \pm 0.491$ | $6.507 \pm 0.535$ | $14.029 \pm 0.795$ | $9.231 \pm 0.258$ |

Table 12: Ablation study on different components of the reaction encoder. Brief results were introduced in Table 4. Numbers in green (yellow) indicate the best (second best) performer.

### E.6. Additional results on ablation study on regularizer

In this section, we present the MAE and RMSE values for the ablation study on regularizer $\text{Reg}(R, I)$ by taking the hyperparameter $\rho = 0$ (Section 5.2), which extends the results of Table 5. From Table 13 summarizes that the addition of the regularizer improves the performance in almost all of the cases.

| | Mean Absolute Error (MAE) | | | | | | |
|---|---|---|---|---|---|---|---|
| **Method** | **GP** | **NS1** | **NS2** | **NS3** | **NS4** | **NS5** | **SC** | **DF** |
| $\rho = 0$ | $23.650 \pm 0.465$ | $9.929 \pm 0.627$ | $8.590 \pm 0.885$ | $8.035 \pm 0.903$ | $7.162 \pm 0.168$ | $4.397 \pm 0.243$ | $8.810 \pm 0.308$ | $8.959 \pm 0.704$ |
| YIELDNET | $23.152 \pm 0.393$ | $9.245 \pm 0.518$ | $8.387 \pm 0.907$ | $7.914 \pm 0.931$ | $7.015 \pm 0.495$ | $4.382 \pm 0.249$ | $8.751 \pm 0.438$ | $6.941 \pm 0.192$ |

| | Root Mean Squared Error (RMSE) | | | | | | |
|---|---|---|---|---|---|---|---|
| **Method** | **GP** | **NS1** | **NS2** | **NS3** | **NS4** | **NS5** | **SC** | **DF** |
| $\rho = 0$ | $39.666 \pm 0.494$ | $13.228 \pm 0.820$ | $10.892 \pm 0.940$ | $9.919 \pm 0.943$ | $9.815 \pm 0.313$ | $6.524 \pm 0.547$ | $13.351 \pm 0.540$ | $11.942 \pm 0.822$ |
| YIELDNET | $38.195 \pm 0.436$ | $12.415 \pm 0.866$ | $10.701 \pm 0.966$ | $9.702 \pm 0.969$ | $9.383 \pm 0.491$ | $6.507 \pm 0.535$ | $14.029 \pm 0.795$ | $9.231 \pm 0.258$ |

Table 13: Ablation study on regularizer $\text{Reg}(R, I)$ in Eq. (21). Brief results were introduced in Table 5. Numbers in green indicate the best performer.

### E.7. Comparison between different ways to compute alignment matrix

Here, we compare an alternative way to construct alignment matrix $P$ mentioned in Eq. (5) by choosing $C[u, v] = h_R(u)^\top h_I(v)$ (referred as $C_{\text{dot}}$) to our chosen one, $C[u, v] = -\sum_{\ell=1}^{d} \max(h_R(u), h_I(v))[\ell]$ (referred as $C_{\text{max}}$). From Table 14 we observe almost similar performances for both the $C$ matrices.

| | Mean Absolute Error (MAE) | | | | | | |
|---|---|---|---|---|---|---|---|
| **Method** | **GP** | **NS1** | **NS2** | **NS3** | **NS4** | **NS5** | **SC** | **DF** |
| $C_{\text{dot}}$ | $22.735 \pm 0.543$ | $8.709 \pm 0.416$ | $8.654 \pm 0.885$ | $8.121 \pm 0.896$ | $6.606 \pm 0.247$ | $4.365 \pm 0.267$ | $8.481 \pm 0.330$ | $6.964 \pm 0.132$ |
| $C_{\text{max}}$ | $23.152 \pm 0.393$ | $9.245 \pm 0.518$ | $8.387 \pm 0.907$ | $7.914 \pm 0.931$ | $7.015 \pm 0.495$ | $4.382 \pm 0.249$ | $8.751 \pm 0.438$ | $6.941 \pm 0.192$ |

| | Root Mean Squared Error (RMSE) | | | | | | |
|---|---|---|---|---|---|---|---|
| **Method** | **GP** | **NS1** | **NS2** | **NS3** | **NS4** | **NS5** | **SC** | **DF** |
| $C_{\text{dot}}$ | $37.639 \pm 0.639$ | $11.459 \pm 0.599$ | $11.010 \pm 0.938$ | $10.002 \pm 0.916$ | $9.167 \pm 0.463$ | $6.514 \pm 0.618$ | $12.984 \pm 0.601$ | $9.322 \pm 0.165$ |
| $C_{\text{max}}$ | $38.195 \pm 0.436$ | $12.415 \pm 0.866$ | $10.701 \pm 0.966$ | $9.702 \pm 0.969$ | $9.383 \pm 0.491$ | $6.507 \pm 0.535$ | $14.029 \pm 0.795$ | $9.231 \pm 0.258$ |

Table 14: Performance of YIELDNET with different ways to calculate $P$ mentioned in Eq. (5)

## E.8. Comparison against competitive baselines with additional datasets

| | | | Mean Absolute Error (MAE) | | | | |
|---|---|---|---|---|---|---|---|
| **Model** | **GP1** | **GP2** | **USPTO** | **DF1** | **DF2** | **DF3** | **NS** |
| GCN | $34.771 \pm 0.399$ | $37.316 \pm 0.309$ | $34.336 \pm 0.345^*$ | $9.610 \pm 0.692^*$ | $10.098 \pm 0.775$ | $10.451 \pm 0.823$ | $6.542 \pm 0.139^*$ |
| HGT | $38.345 \pm 0.241$ | $41.798 \pm 0.276$ | $34.767 \pm 0.320$ | $11.246 \pm 0.843$ | $10.721 \pm 0.889$ | $10.989 \pm 0.916$ | $8.608 \pm 0.218$ |
| TAG | $36.504 \pm 0.292$ | $39.757 \pm 0.239$ | $34.693 \pm 0.321$ | $11.200 \pm 0.774$ | $10.553 \pm 0.919$ | $10.877 \pm 0.919$ | $9.083 \pm 0.132$ |
| GIN | $36.184 \pm 0.184$ | $39.008 \pm 0.280$ | $34.896 \pm 0.304$ | $11.169 \pm 0.839$ | $10.373 \pm 0.871$ | $10.610 \pm 0.805$ | $8.883 \pm 0.155$ |
| DeepReac+ | $29.700 \pm 0.210$ | $32.577 \pm 0.402$ | $36.890 \pm 0.994$ | $12.302 \pm 1.693^*$ | $10.665 \pm 0.928$ | $10.091 \pm 0.689$ | $9.450 \pm 0.905$ |
| YieldBERT | $39.587 \pm 0.278$ | $44.072 \pm 0.280$ | $35.352 \pm 0.356$ | $10.126 \pm 0.822^*$ | $9.912 \pm 0.925$ | $10.475 \pm 0.656$ | $7.837 \pm 0.154$ |
| YIELDNET | $24.269 \pm 0.610$ | $27.563 \pm 0.618$ | $33.460 \pm 0.474$ | $8.832 \pm 0.747$ | $8.552 \pm 0.894$ | $8.815 \pm 0.384$ | $6.278 \pm 0.194$ |

| | | | Root Mean Squared Error (RMSE) | | | | |
|---|---|---|---|---|---|---|---|
| **Model** | **GP1** | **GP2** | **USPTO** | **DF1** | **DF2** | **DF3** | **NS** |
| GCN | $40.945 \pm 0.347$ | $42.132 \pm 0.300^*$ | $38.257 \pm 0.383^*$ | $13.165 \pm 0.963^*$ | $12.065 \pm 0.688$ | $13.523 \pm 0.842$ | $9.055 \pm 0.214^*$ |
| HGT | $42.667 \pm 0.209$ | $44.633 \pm 0.239$ | $37.913 \pm 0.374^*$ | $14.38 \pm 1.072$ | $12.575 \pm 0.819$ | $14.172 \pm 0.836$ | $11.454 \pm 0.251$ |
| TAG | $41.551 \pm 0.221$ | $43.536 \pm 0.317^*$ | $37.970 \pm 0.397^*$ | $14.274 \pm 1.011$ | $12.602 \pm 0.831$ | $14.054 \pm 0.835$ | $11.974 \pm 0.185$ |
| GIN | $41.836 \pm 0.226$ | $42.983 \pm 0.257^*$ | $38.006 \pm 0.385^*$ | $14.399 \pm 1.071$ | $12.296 \pm 0.784$ | $13.741 \pm 0.748$ | $11.764 \pm 0.225$ |
| DeepReac+ | $37.576 \pm 0.241^*$ | $40.961 \pm 0.371$ | $42.876 \pm 1.157^*$ | $15.618 \pm 1.724^*$ | $13.136 \pm 0.942$ | $12.902 \pm 0.839^*$ | $12.028 \pm 0.917$ |
| YieldBERT | $44.099 \pm 0.165$ | $46.380 \pm 0.220$ | $38.446 \pm 0.396^*$ | $13.669 \pm 1.146^*$ | $12.110 \pm 0.822$ | $13.529 \pm 0.726$ | $10.678 \pm 0.217$ |
| YIELDNET | $38.862 \pm 0.622$ | $42.852 \pm 0.659$ | $41.191 \pm 1.112$ | $12.181 \pm 0.903$ | $10.790 \pm 0.865$ | $11.813 \pm 0.478$ | $8.777 \pm 0.310$ |

Table 15: Comparison of yield prediction performance for YIELDNET against all the competitive baselines, *viz.*, GCN (Kipf & Welling, 2017), HGT (Hu et al., 2020), TAG (Du et al., 2017), GIN (Xu et al.), DeepReac+ (Gong et al., 2021), YieldBERT (Schwaller et al., 2021c), on the 20% test examples, across the additional datasets. Performance is measured in terms of MAE (top half) and RMSE (bottom half). Numbers in green (yellow) indicate the best (second best) performer. Numbers with $^*$ indicate that either the model outperforms YIELDNET in that case or the difference is not statistically significant (i.e. p-value $> 0.05$). The results reveal the same observations as in Table 1.

In this section, we present the MAE and RMSE for all the methods on the additional datasets *viz.*, NS4, NS5, DF1, DF2, DF3, and NS. Table 15 summarizes the results, which shows that our method outperforms the baselines, similar to Table 1. For the rest of the datasets YIELDNET outperforms the baselines by a significant margin in most cases. We observe the results are more prominent in the case of MAE as compared to RMSE.

### E.9. Results on approximating the gold permutation for additional datasets

We compute $||\boldsymbol{P} - \boldsymbol{P}^*||_F$ (mentioned in Section 5.2) for attention and YIELDNET for two additional GP1 and GP2 datasets

| Model | GP | GP1 | GP2 |
|-------|-----|------|------|
| Attention | 5.096 | 4.611 | 4.496 |
| YIELDNET | 3.708 | 3.627 | 3.165 |

Table 16: Comparison of $||\boldsymbol{P} - \boldsymbol{P}^*||_F$.

as true labels of permutations for Gas-phase reaction datasets are available. Table 16 summarizes that the atom mapping approximation in YIELDNET is superior to that of attention.

### E.10. Results on the interplay between the quality of learned alignment and predicted yield for additional datasets

We plot $\mathbb{E}(\Delta \mathrm{AE}_r | \Delta \boldsymbol{P}_r)$ vs $\Delta \boldsymbol{P}_r$ same as Figure 6 for additional GP1 and GP2 datasets in Figure 7. This shows a similar trend i.e. high correlation between error in yield prediction and atom-to-atom alignment for GP1 and GP2 datasets.

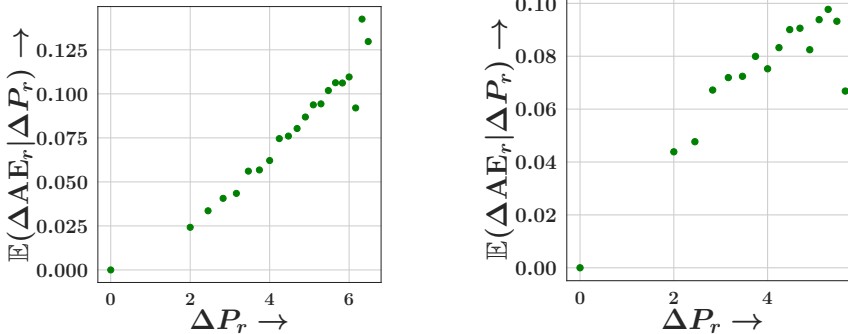

Figure 7: $\mathbb{E}(\Delta \mathrm{AE}_r | \Delta \boldsymbol{P}_r)$ vs $\Delta \boldsymbol{P}_r$ plot for **GP1** (left) and **GP2** (right) dataset

### E.11. Visualization of condensed graphs where they show the exact transition states

Condensed graphs serve as surrogates of the transition states. We are visualizing some cases for the GP dataset, where they reflect the exact transition state. We select the reactions with low MAE values and collect the alignment matrix, $P$ corresponding to those reactions. Then, we apply the Hungarian algorithm to get hard permutation matrices, $P_{hard}$. We obtain the $\text{Adj}_{CGR}$ as follows, $\text{Adj}_{CGR} = \max(\text{Adj}_R, P_{hard,r}\text{Adj}_I P_{hard,r}^\top)$. Finally from the connections obtained from the $\text{Adj}_{CGR}$, we draw the molecular structures of condensed graphs as shown in Figure 8.

Figure 8: Representative examples of the condensed graphs where they show the exact transition states, (shown on the top of the arrow in each case) as generated by YIELDNET during yield prediction.

We use NetworkX (Hagberg et al., 2008) to get the connections between atoms from $\text{Adj}_{CGR}$. After obtaining those NetworkX raw graphs we manually draw figures of Figure 8 in ChemDraw for better visualization, maintaining the connectivity obtained through the NetworkX graphs.

### E.12. Time complexity

We also report the inference time for the NS3 dataset with the test size of 60 and batch size of 8. From Table 17, it is clear that inference time for our model is comparable with the some of the GNN based models (HGT and TAG) and not as high as YieldBERT.

| Model | Inference time in sec |
|---|---|
| GCN | $1.727 \pm 0.010$ |
| HGT | $4.684 \pm 0.087$ |
| TAG | $3.109 \pm 0.023$ |
| GIN | $1.693 \pm 0.020$ |
| DeepReac+ | $0.900 \pm 0.026$ |
| YieldBERT | $84.983 \pm 0.186$ |
| YIELDNET | $3.440 \pm 0.298$ |

Table 17: Inference time for different methods

