# OpenReview forum: "Learning Condensed Graph via Differentiable Atom Mapping for Reaction Yield Prediction"
_ICML.cc/2025/Conference — ICML 2025 poster_

### Official Review · Reviewer_TZPS · 2025-02-22

**Overall Recommendation:** 3

**Summary:**

The paper introduces YIELDNET, a neural model designed to predict chemical reaction yields by learning a condensed graph representation of reactions. Unlike traditional methods that rely on quantum chemistry-based molecular descriptors, YIELDNET approximates atom mapping—the correspondence between reactant and product atoms—using a differentiable node alignment network. This mapping allows the construction of a Condensed Graph of Reaction (CGR), a supergraph that serves as a surrogate for the transition state, a crucial determinant of reaction yield. The model then processes CGR embeddings through a transformer-guided reaction path encoder to predict yields. A key advantage of YIELDNET is that it operates under distant supervision, meaning it does not require explicit atom mapping or transition state labels during training, yet it outperforms baseline models. This approach enhances inductive bias by integrating a differentiable approximation of the transition state, improving the accuracy of yield predictions in multi-step chemical reactions.

**Claims And Evidence:**

Several claims are stated in this paper, including:

1) The CGR serves as a surrogate for the TS -- The claim is not rigorously validated. The claim relies on structural similarity but ignores electronic structure and activation energy, which are critical in TS determination. Transition states require potential energy surface calculations, which the authors do not provide. A surrogate for the TS must be benchmarked against computed TS structures to be credible. Even though in Figure 8 the author shows how TS is represented, for more complicated reactions, the middle state can also be intermediate. This simplicity of the whole reaction scheme should be emphasized.

2) The differentiable atom mapping is an effective approximation of the true atom mapping. The authors state that YIELDNET approximates atom mapping via a differentiable node alignment network (using Sinkhorn iterations). Experiments proved that the model outputs a doubly stochastic alignment matrix that relaxes hard permutation constraints, which mitigates NP-hard graph matching challenges. However, the improvement compared to other mapping methods is marginal. Test cases where the mapping fails are not mentioned —does it work for rearrangements, symmetry-driven reactions, and pericyclic processes? What about other edge cases (e.g., radical shifts, metal-catalyzed bond formations)?

3) The author claims that yield prediction can be learned w/o explicit TS information, giving experiments of empirical comparison with existing ML models for yield prediction. But thermodynamically, the yield is decided by: 1) reaction duration + activation energy height; 2) activation energy barriers comparison of multiple reaction pathways that compete with each other. The interpretation of the inherent state does not guarantee any physical understanding or explanation of its performance.

4) The author states that YIELDNET generalizes well across different reaction types
.

**Essential References Not Discussed:**

Related and up-to-date papers are properly cited.

**Experimental Designs Or Analyses:**

The experimental design and analysis in YIELDNET focus on reaction yield prediction using differential atom mapping, and condensed reaction graphs (CGRs). While the methodology is innovative, the experimental design has limitations in terms of chemical validity, dataset composition, benchmarking, and mechanistic interpretability.

The paper claims to predict reaction yields by modeling CGR embeddings and atom mappings, but it does not explicitly explain the relationship between TSs defined by the energy barrier and the connectivity graphs used in their model. Besides, other terms that will influence yield (reaction time, solvent, catalysts...) are ignored. Since yield is highly condition-dependent, the author should consider design and analyze the limitations.

The authors collected several datasets for the training and test of YIELDNET, but it lack detailed yield distribution visualization and reaction introductions to help readers understand the difference between the dataset and the potential bias involved.

**Methods And Evaluation Criteria:**

The author performs proper methods to design the model and organized a reaction dataset for the problem.

**Other Comments Or Suggestions:**

N/A

**Other Strengths And Weaknesses:**

Strengths: The introduction of differentiable atom mapping for reaction yield prediction is an innovative step. Unlike traditional rule-based or SMILES-based atom mapping methods, YIELDNET relaxes atom mappings into a soft differentiable space using the Sinkhorn algorithm.

Weakness: The model is only tested on a limited dataset and the generalization ability of the model is still a mystery. Besides, The model does not analyze when and why YIELDNET fails and how missing reaction condition embeddings influence the performance.

**Questions For Authors:**

1. How does the Condensed Graph of Reaction (CGR) meaningfully approximate the transition state? The paper claims that the CGR serves as a surrogate for the transition state (TS), but it does not include quantum mechanical validation of this claim. If I can surrogate the "intermediate" with 0.8*R+0.2*P; 0.5*R+0.5*P,0.2*R+0.8*P would that be helpful?

2. How does YIELDNET compare to quantum-ML hybrid models (with QM-optimized TS) for yield prediction? If it does lend the TS information, there should be similarities.

3. How does YIELDNET generalize across different reaction classes?

4. Why does YIELDNET ignore reaction conditions (temperature, solvent, catalyst effects)? How you would design the model to incorporate this situation as the condensed graph may introduce a catalyst as a part of the reagent during a specific step of the reaction?

5. What are the major failure modes of YIELDNET?

**Relation To Broader Scientific Literature:**

YIELDNET could be a potentially useful model for yield prediction. However, the paper still lacks sufficient connection to fundamental chemistry, including physical organic chemistry principles such as transition state theory, kinetic control, and thermodynamics. To make it a more persuasive model with chemistry understanding, the author should report the validation of model generalization on unseen reaction classes.

**Theoretical Claims:**

N/A

---

> ### Author Rebuttal · Authors · 2025-04-01
>
> We thank the reviewer for their comments, which we address as follows.
>
> > *CGR meaningfully approximates TS?*
>
> During rebuttal, we performed quantum chemical validation analogous to that reported by Choi, in 'Prediction of transition state structures of gas-phase chemical reactions via machine learning'  Nature Commun. 2023.
>
> Here, we have used our CGR based TS as the initial guess for quantum chemical optimization for reactions in a GP dataset. We observed that 80% of these converged to TS. Subsequent QM calculations could produce correct reactant and product from the TS for 60% cases.
>
> In addition, we also compare CGR-based TSs with true (QM-computed) TSs using  two metrics:
>
> (1) Accuracy: The fraction of total connections in the CGR-based TS which match with the  QM computed TS.
> (2) RMSD:  Root mean squared deviation (RMSD -lower the better) between atom positions obtained from CGR-based TS and the QM computed TS. Results for the GP dataset for our method and other interpolation based methods as follows. We observe that our method performs better.
>
>  ||Accuracy |RMSD|
>  |-|-|-|
>  |0.8R+0.2P|0.51|1.50|
>  |0.5R+0.5P|0.71|1.24|
>  |0.2R+0.8P|0.91|1.33|
>  |Our|**0.95**|**0.71**|
>
>
> Our CGR serves as a surrogate for TS to enhance yield prediction, not to provide an accurate TS prediction.  More accurate TS prediction requires TS supervision during training, which most yield datasets lack.  Complex datasets like SC contain ~200 atoms per reaction. Here, QM calculations on TSs take a couple of days per reaction--- assuming a good initial TS guess, which is even more time-consuming given the complexity of the dataset. Hence, general evaluation for our CGR was infeasible.
>
> However, as the reviewer suggested, we performed experiments on reaction instances in the GP dataset, which shows that despite absence of supervision, our CGR shows better proximity to the TS than its alternatives.
>
> > *compare to quantum-ML models*
>
> We  feed the QM-optimized-TS (TS*) in our model to compute yield prediction error $E^*(r) = |y^*(r)-y _{TS^*}(r)|$ for each reaction $r$ and then compute yield prediction error $E(r) = |y^*(r)-y _{CGR}(r)|$ provided by our CGR based TS. The average difference between these errors is $<$ 8% of QM-optimized-TS based yield prediction error.
>
> > *generalize to different reaction classes*
>
> We compute MAE on test examples that include reactions from SC dataset, whereas the model is trained on DF dataset. We observe that our method shows better performance.
>
> |||
> |-|-|
> |GCN|13.40|
> |HGT|25.38|
> |TAG|18.52|
> |GIN|17.03|
> |YB |23.64 |
> |Our |10.60 |
>
> > *ignore reaction conditions?*
>
> We do incorporate both catalyst and solvent, when present, into the reactant set R. We will explicitly mention them in the revised paper. Temperature was excluded as it is invariant in the case of high-throughput experimentation datasets (e.g., DF and NS datasets), which maintain consistent reaction conditions throughout, or varies mildly for others (e.g., SC). Integrating temperature into the node/edge features might enhance CGR or yield prediction quality.
>
> > *a catalyst as a part of the reagent*
>
> In i-th step of a multi-step reaction,  the reactant set $R_i$ is a disjoint graph of all the participating species including catalyst, base, solvent, etc.
>
> >*Failure mode*
>
> Upon close inspection, we observe that our method couldn't do well in reactions that are very unusual/rare in terms of bond formation and breaking, due to their under-representation in training samples. Poor predictions on such  reactions are unlikely to make any adverse impact, since they are too rare. If the dataset labels (yield values) carry large measurement errors, then also our model (and the baselines) would perform poorly.
>
> >*lack yield visualization*
>
> yhist.pdf in https://bit.ly/rbynet  shows that our predicted yield distribution mimics true yield distribution, very closely.
>
> >*yield for rearrangements, pericyclic processes*
>
> Our datasets include rearrangements (present in GP), pericyclic processes (present in GP), metal-catalyzed bond formations (all reactions in SC), etc.
> Following are MAE based comparisons.
>
> |-|Our|Closest baseline|
> |-|-|-|
> |rearrangement|21.82|DeepReac+: 28.79|
> |Pericyclic|23.26|GCN: 24.16|
> |Metal-catal.|8.75|DeepReac+: 10.44|
>
> > *experimental design has limitations, dataset, benchmarking*
>
> We evaluated over 10 SOTA datasets against six baselines across ten splits for all experiments, including ablation study, to maintain consistency. We believe that our work conducts experiments with more rigor and comprehensiveness than any previous yield prediction baselines.
>
> We found that the pre-final layer embedding $z_r$ (Eq. 19) correlates with activation energies (corr.pdf in https://bit.ly/rbynet). Extracting mechanistic signals from yield alone is challenging and no prior yield prediction work attempts it-- they only focus on black box yield predictors. Hence, our work makes notable progress, paving the way for future research.

---

### Official Review · Reviewer_Vr41 · 2025-03-13

**Overall Recommendation:** 4

**Summary:**

Predict permutation-matrix of a chemical reaction with GNN atom embeddings and iterative Sinkhorn interactions. Sampling from the permutation-matrix to get the atom-mapping which is used by a Transformer to predict the yield.

**Claims And Evidence:**

- propose differentiable approximation of the atom-mapping: I would argue that e.g. RXNMapper is also a "differentiable approach to atom mapping" but in their evaluation they show that they outperform RXNMapper (Table 2). Here also +- standard deviation as in Table 1 would be nice as well as statistical significancance tests (further state which test, and assumtion has been used). Their approach is novel and performant but seems memory intensive.
- proposed YieldNet outperforms several baselines: experimentally validated in Table 1; code and data provided to reproduce as supp. across multiple datasets via 10-fold cross-validation; (why is USPTO in the supp but not the main Table 1?)

**Essential References Not Discussed:**

OK

**Experimental Designs Or Analyses:**

All evaluations are sound.
Further evaluations on different splits might be interesting but overall the analysis is done in a just manner.

**Methods And Evaluation Criteria:**

yes

**Other Comments Or Suggestions:**

Suggestions to add to clarity:
- you mention several times that activation energy directly impacts yield; what is the precise relationship
- it seems that the topic of yield in reality is much more nuanced then described in the manuscript: e.g. conditions such as person executing the experiment, temperature, .... I guess it's fair to distinguish between theoretical idealized yield and practical yield.
- For reaction-path encoder - how is the atom mapping encoded here / how does it find it's way to the final yield? it seems the H_CGR is an embedding that holds this information for one step --> why not incorporate the whole reaction and perform attention over that

**Other Strengths And Weaknesses:**

Original - yes
Significance - yes
Clarity - yes - paper is very well explained (minor suggestions follow)

**Questions For Authors:**

no further questions

**Relation To Broader Scientific Literature:**

Key contributions could be demarcated better from scientific literature.

**Theoretical Claims:**

Not checked.

---

> ### Author Rebuttal · Authors · 2025-04-01
>
> We thank the reviewer for their comments, which we address below.
>
> > *results for atom mapping*
>
> Appendix E.3 contains the table with +- std error. Paired t-test in majority of cases revealed that the performance gain achieved by our method is statistically significant with $p=0.01$
>
> > *Approach is novel but memory intensive*
>
> Our method utilizes the most compute on computing the permutation matrix using Sinkhorn iterations, which has $O(N^2)$ complexity if $N$ is the total number of nodes. However, this is not a hard bottleneck and can be easily overcome using low rank OT.
>
> Low rank OT provides highly efficient Sinkhorn iterations (see Scetborn et al. Low-Rank Sinkhorn Factorization, ICML 2021). We first apporximate $M = \sum _{i=1} ^d  \max (H  _R[u,i], H  _I [u',i])\approx AB^T$ where $A$ and $B$ are $N \times d$ and $N \times d$ low rank matrices. Similarly, we approximate the transport matrix $P$ as $P \approx Q D(1/g) R^T$ where $Q, R$ are $N \times r$ matrices, $g$ is $r \times 1$, and $D(1/g)$ is a diagonal matrix with entries $1/g_i$. To ensure $P$ is doubly stochastic, we enforce:
>
> $$
> Q\mathbf{1} = \mathbf{1}, \quad \mathbf{1}^T Q = g^T
> $$
>
> $$
> R\mathbf{1} = \mathbf{1}, \quad \mathbf{1}^T R = g^T
> $$
>
> $$
> g^T \mathbf{1} = 1, \quad Q, R, g > 0
> $$
>
> Instead of minimizing $Tr(P^T M)$, we minimize:   $Tr((Q D(1/g) R^T)^T AB^T)$ using alterting minimization wrt $Q, R, g$. To optimize $Q$, we rewrite:
> $$
> Tr((Q D(1/g) R^T)^T AB^T) = \langle Q, AB^T R D(1/g) \rangle
> $$
> We apply Sinkhorn iterations to: $X = AB^T R D(1/g)$, computed in 3 steps. Note that complexity  of multiplying $m$ x $n$ and $n$ x $p$ matrices is $O(mnp)$
>
> 1. Compute $R D(1/g)$, which is $O(Nr)$ since $R$ is $N \times r$, $g$ is $r \times  1$.
> 2. Compute  $B^T R D(1/g)$ which is $O(dNr)$ since $B^T: d \times N$ and  $R D(1/g): N\times r$
> 3. Compute  $X = AB^T R D(1/g)$ which is $O(Ndr)$  since $A: N \times d$ and  $B^T R D(1/g): d\times r$
>
> Thus, complexity reduces from $O(N^2)$ to $O(Ndr)$. The same efficiency gain applies when optimizing $R$ and $g$.
>
> Moreover, in terms of time efficiency,   our model is comparable with the some of the GNN based models (HGT and TAG) but not as high as YieldBERT.
>
> |Model|time (sec)|
> |-|-|
> |GCN|1.73|
> |HGT|4.68|
> |TAG|3.11|
> |GIN|1.69|
> |DeepReac+|0.9|
> |YieldBERT|84.98|
> |YIELDNET|3.44|
>
>
> > *USPTO ...not in the main Table 1?*
>
> We agree with the reviewer. We would revise Table 1 to include the USPTO results in the main text.
>
>
> > *....activation energy directly impacts yield; what is the precise relationship*
>
> For a single-step reaction assuming first order kinetics, the relationship is as follows.
> Suppose,  $\Delta G^\ddagger$ represents activation energy in Jmol$^{-1}$.   $k_B, h, R$ are the Boltzmann constant ($1.38\times 10^{-23}$ JK$^{-1}$), Planck constant ($6.626\times 10^{-34}$Js), and universal gas constant ($8.314$ JKmol$^{-1}$) respectively. Here, $t$ is the reaction time in sec. If "Temp" is the reaction temperature, then $k$, the first order rate constant is:
> $$  k = \dfrac{k_B \text{Temp}}{h} \exp\left(\dfrac{-\Delta G^{\ddagger}}{R\cdot\text{Temp}}\right),$$
>  Yield is given by:
> $$yield = (1 - \exp(-k t)) \times 100$$
>
>
>
>
>
>
>
> > *conditions such as person executing the experiment, temperature, etc. .... I guess it's fair to distinguish between theoretical idealized yield and practical yield.*
>
> There are several factors that could influence yields. We do incorporate both catalyst and solvent, when present, into the reactant set R.   Temperature was excluded as it is invariant in the case of high-throughput experimentation datasets (e.g., DF and NS datasets), which maintain consistent reaction conditions throughout, or varies mildly for others (e.g., SC). Other variabilities (e.g., purification procedure) are lacking in the state-of-the-art datasets, which is why we were unable not consider them. We discussed these factors in the conclusion. We will elaborate them further, if our paper gets accepted.
>
>
>
> > *why not incorporate the whole reaction and perform attention over that*
>
> Yes, $H_{CGR}$ captures the signals from atom mapping. We experimented with the transformer model where we incorporate whole reaction and perform attention over all reaction components, without explicit CGR modeling.  Following MAE values show that our method performs significantly better.
>
> | Method|NS1|NS2|NS3|SC|
> |-|-:|-:|-:|-:|
> |Whole-reaction-attention|11.356|9.459|9.342|10.686|
> |**YIELDNET**|**9.245**|**8.387**|**7.914**|**8.751**|
>
> Despite the increased model size, whole-reaction-attention performs worse. We believe this is due to the fact that attention is non-injective in nature, i.e., multiple atoms from reactants can be mapped one single atom in product. In contrast, permutation is injective-- they provide one-to-one mapping between atoms. One way to mitigate the problem of transformer  is to design permutation induced transformer, which we believe has a strong potential in this direction.

---

> > ### Comment · Reviewer_Vr41 · 2025-04-03
> >
> > Thank you to the authors for providing a detailed and comprehensive rebuttal. I appreciate the effort taken to address the questions raised in my review.
> >
> > The additional clarifications, analyses, and justifications provided were helpful and have certainly added clarity regarding the work presented.
> >
> > While these clarifications are valuable and appreciated, my overall assessment based on the points discussed leads me to maintain my current score.
> >
> > (If only OpenReview allowed for "comma point" increases – the added clarity certainly deserves acknowledgement, even if it doesn't shift the final recommendation!)

---

> > > ### Author Response · Authors · 2025-04-09
> > >
> > > We would like to thank the reviewer for their positive review and encouraging comments.

---

### Official Review · Reviewer_brLH · 2025-03-14

**Overall Recommendation:** 3

**Summary:**

The paper introduces YIELDNET, a neural yield prediction model designed to predict the yield of multi-step chemical reactions without explicit atom mapping supervision. The key contributions include a differentiable node alignment network to approximate atom mapping, the construction of a CGR as a surrogate for transition states, and a transformer-guided reaction path encoder to model multi-step reactions. The approach enables end-to-end learning with supervision only from yield values, outperforming all baselines across eight datasets.

**Claims And Evidence:**

The paper claims that YIELDNET can predict reaction yields more accurately than existing methods by leveraging an approximate transition state representation. This claim is supported by experimental comparisons showing significant improvements over baselines. The ablation studies on different components of the model, such as CGR representations, reaction path encoder components, and the regularizer, provide evidence for the importance of these elements in the model.

**Essential References Not Discussed:**

No

**Experimental Designs Or Analyses:**

The experimental setup is sound, with multiple datasets and comparisons against reasonable baselines. The paper provides quantitative evidence of YIELDNET’s superiority. The ablation studies are well designed to analyze the impact of different components of the model

**Methods And Evaluation Criteria:**

The proposed methods make sense for the problem. The differentiable node alignment network is a novel approach to approximate atom mapping, which is a crucial step in the reaction yield prediction problem. The evaluation criteria, including using MAE and RMSE to measure the performance on test sets, are standard and reasonable for this type of prediction task. The datasets cover a variety of reaction types for evaluating the model's performance.

**Other Comments Or Suggestions:**

No

**Other Strengths And Weaknesses:**

Strengths: 1. The combination of differentiable atom mapping, CGR approximation, and reaction path encoding in a single model for yield prediction is novel.
2. Strong empirical performance across multiple datasets.
3. The work has practical significance as accurate yield prediction can help in chemical synthesis design and optimization.
Weaknesses: The method may face challenges when applied to larger molecular reactions due to the O(N^{2}) complexity of computing the alignment matrix P. Although the authors mention a possible mitigation strategy, it needs further verification.

**Questions For Authors:**

1.What is the computational overhead of this method compared to baselines? The Sinkhorn operator involves iterative row and column normalization over a cost matrix. Given an N×N permutation matrix, the complexity is approximately O (T N^2), where T is the number of Sinkhorn iterations. Additionally, the gradient computation for backpropagation through Sinkhorn updates introduces further memory and compute costs. For a multi-step reaction with n steps, the embeddings from each step are concatenated and passed through a transformer.
2.Would a lighter-weight model maintain accuracy while improving efficiency?

**Relation To Broader Scientific Literature:**

The key contributions of the paper are closely related to the broader scientific literature. In the area of reaction yield prediction, previous works mainly rely on quantum chemically computed molecular descriptors, molecular fingerprints, or SMILES based representation. YIELDNET differentiates itself by leveraging graph neural networks and approximating atom mapping and transition states.

**Theoretical Claims:**

The paper does not present complex theoretical proofs. The design of the node alignment network is based on relaxation of a quadratic assignment problem (QAP) to a linear optimal transport (OT) problem. The authors claim that the alignment matrix obtained is the solution of the entropy regularized linear OT problem. While the theoretical background is well presented, it would be beneficial to have a more detailed proof or reference to a more in-depth theoretical analysis to ensure the correctness of this claim.

---

> ### Author Rebuttal · Authors · 2025-04-01
>
> We thank the reviewer for their suggestions, which we address as follows.
>
> > *theoretical analysis*
>
> We provide the following theoretical underpinning. We start with the QAP in Eq 8. If $\mathcal{P}$ is the set of permutations,  then QAP is
> $$\min  _{P\in \mathcal{P}}
> \underbrace{\sum _{u,v}  \max (Adj  _R,P Adj _I P^{\top}) [u,v]} _{:=c(R,I;P)} --- (QAP)
> $$
> The standard way to minimize it is to use Gromov Wasserstein (GW) based projection [A1], where P is updated as
>
> $$    P  \leftarrow  \text{argmin}  _{P} \textrm{Trace}\left(P^T\nabla  _{P}\ c(R,I;P) \right)  -- (grad) $$
> In our work, we provide a neural approximation of the gradient $\nabla  _{P}\ c(R,I;P) \approx [\sum  _{i=1} ^d  \max (H  _R[u,i], H
>  _I [u',i])] _{u,u'}$. We use max to keep  aligned with max() in Eq (QAP). This approximates Eq. (grad) as the following linear OT (Eq 9):
>
> $$ \min  _{P\in \mathcal{P}}  \underbrace{ \sum  _{u,u'}  \sum _{i=1} ^d  \max(H  _R[u,i], H  _I [u',i]) P _{uu'}} _{F(P)}  -- (a) $$
>
> Eq (a) is optimization over permutations, which may appear to be computationally hard. However, we can write it as an instance of the following optimization over the space of doubly stochastic matrices $B = (P: P   \ge 0,  P\mathbf{1} = P^T\mathbf{1} = 1)$.
>
> $$ \min _{P}F(P),  \text{  s.t. }  P\in B --- (b)  $$
>
> Eq (b) is equivalent to Eq (a) since it is an linear program.The optimal solution of a linear program always coincides with some vertex of the space induced by the set of linear constraints. Thus, the optimal solution of Eq (b) is a permutation matrix $P^*$, as the set of permutation matrices are the vertices of B, the space   of doubly stochastic matrices.
>
> However, $\arg\min _{P}F(P)$ in Eq (b) is non-differentiable. To enable differentiation, we approximate it using Sinkhorn iterations, analogous to how argmax is approximated using softmax. Given n numbers $G_1,..G_n, \ \ \mathbb{1}[i=\arg \min _{j} G_j] \approx p_i=\frac{e^{-G_i/\lambda}}{\sum _{j}e^{-G _{j}/\lambda}}$, where $\mathbb{1}$ is indicator function. One can show that $p_i: i\in [n]$  minimizes the following entropy regularized optimization problem:
>
> $$\min _{p} \sum _{j} p _j G _j +\lambda. \sum _{j} p _j \log(p _j),\text{ such that } 0 \le p _j \le 1, \sum _{j} p _j =1$$
>
> Using the same technique, we can show that Sinkhorn mechanism minimizes the entropy regularized linear OT problem $\min _P F(P) - \lambda \text{Entropy}(P)$ [A2].
>
> [A1] Xu et al. Gromov-wasserstein learning for graph matching and node embedding, ICML 2019
>
> [A2] Mena et al. Learning Latent Permutations with Gumbel-Sinkhorn Networks. ICLR 2018.
>
> > *Computational overhead*
>
> Following table shows the computational overhead in terms of inference time. It shows: our model is comparable with the some of the GNN based models (HGT, TAG) and not as high as YieldBERT.
>
> |Model|time (sec)|
> |-|-|
> |GCN|1.73|
> |HGT|4.68|
> |TAG|3.11|
> |GIN|1.69|
> |DeepReac+|0.9|
> |YieldBERT|84.98|
> |YIELDNET|3.44|
>
> Moreover, the training time per epoch for our model is 8 seconds (s), which is comparable with baselines: 6s for HGT, 10s for YieldBERT (with same batch size)
>
> *Efficiency enhancement:* Low rank OT provides highly efficient Sinkhorn iterations. See Scetborn et al. Low-Rank Sinkhorn Factorization, ICML 2021. One first apporximates $M = \sum _{i=1} ^d  \max (H  _R[u,i], H  _I [u',i])\approx AB^T$ where $A$ and $B$ are $N \times d$ and $N \times d$ low rank matrices. Similarly, we approximate the transport matrix $P$ as $P \approx Q D(1/g) R^T$ where $Q, R$ are $N \times r$ matrices, $g$ is $r \times 1$, and $D(1/g)$ is a diagonal matrix with entries $1/g_i$. To ensure $P$ is doubly stochastic, we enforce:
>
> $$Q\mathbf{1} = \mathbf{1}, \quad \mathbf{1}^T Q = g^T$$
>
> $$R\mathbf{1} = \mathbf{1}, \quad \mathbf{1}^T R = g^T$$
>
> $$g^T \mathbf{1} = 1, \quad Q, R, g > 0$$
>
> Instead of minimizing $Tr(P^T M)$, we minimize:  $Tr((Q D(1/g) R^T)^T AB^T)$ using alternating minimization wrt $Q, R, g$. To optimize $Q$, we rewrite:
> $$
> Tr((Q D(1/g) R^T)^T AB^T) = \langle Q, AB^T R D(1/g) \rangle
> $$
> We apply Sinkhorn iterations to: $X = AB^T R D(1/g)$, computed in 3 steps. Note that complexity of multiplying $m \times n$ and $n \times p$ matrices is $O(mnp)$
>
> 1. Compute $R D(1/g)$, which is $O(Nr)$ since $R$ is $N \times r$, $g$ is $r \times  1$.
> 2. Compute $B^T R D(1/g)$ which is $O(dNr)$ since $B^T: d \times N$ and  $R D(1/g): N\times r$
> 3. Compute $X = AB^T R D(1/g)$ which is $O(Ndr)$  since $A: N \times d$ and  $B^T R D(1/g): d\times r$
>
> Thus, complexity reduces from $O(N^2)$ to $O(Ndr)$. The same efficiency gain applies when optimizing $R$ and $g$.
>
> > *lighter model*
>
> We replaced some complex components with simpler components in our model. MAE numbers are as follows.
>
> ||NS1|NS2|NS3|
> |-|-|-|-|
> |step-Aggr=SumAggr|10.19|8.86|8.35|
> |No-Transformer|9.47|8.68|8.16|
> |Our|9.25|8.39|7.91|
>
> We observe that there is some drop in performance. Depending on available resources, one can decrease the complexity to build a lighter model.

---

### Decision · Program_Chairs · 2025-05-01

**Decision:**

Accept (poster)

**Comment:**

This work investigates the problem of predicting yield for (potentially multi-step) reactions, based on structures of the molecules involved alone (without transition state or atom mapping information). This is approached by designing a model which predicts a soft matching between reactant and product atoms, then uses it to create a surrogate for the transition state, and finally processes that to predict yield. Authors show that training this end-to-end with only yield supervision leads to accurate predictions and even learning to predict atom mapping as a by-product.

After the rebuttal period, all three reviewers were voting to accept this paper, with varying degrees of confidence. On the positive side, reviewers praised sensible model design (which draws upon assumptions appropriate for the domain, e.g. transition state influencing yield), strong empirical performance across datasets, and real-world significance (as yield prediction is an important problem in chemistry and drug discovery). On the negative side, some reviewers were initially worried about resource requirements (both time and memory) and generalization; however, these concerns were well-addressed by authors during the discussion period.

Having read the manuscript in detail, I agree with the unanimous reviewer decision: this paper is a solid contribution to ICML. It reads very well, and shows a thoughtfully selected array of experiments and ablations. I recommend acceptance (of medium strength). My main rationale for not going for a strong accept recommendation was that this work may have limited applicability outside of yield prediction (and even there, it's only applicable to reactions with fully specified side products), thus expected audience may be slightly smaller than for an average accepted paper.